PREPARED FOR SUBMISSION TO JHEP

QMUL-PH-22-10, SAGEX-22-23-E

# Radiation, entanglement and islands from a boundary local quench

**Lorenzo Bianchi**[1,2], **Stefano De Angelis**[3], **Marco Meineri**[4]

[1] *Universitá di Torino, Dipartimento di Fisica, Via P. Giuria 1, I-10125 Torino, Italy*

[2] *INFN. - sezione di Torino, Via P. Giuria 1, I-10125 Torino, Italy*

[3] *Centre for Theoretical Physics, Department of Physics and Astronomy, Queen Mary University of London, Mile End Road, London E1 4NS, United Kingdom*

[4] *Department of Theoretical Physics, University of Geneva, 24 quai Ernest-Ansermet, 1211 Genève 4, Suisse*

*E-mail:* lorenzo.bianchi@unito.it, stefano_deangelis@outlook.com, marco.meineri@gmail.com

ABSTRACT: We study the entanglement and the energy density of the radiation emitted after a local quench in a boundary conformal field theory. We use the operator product expansion (OPE) to predict the early- and late-time behavior of the entanglement entropy and we find, under mild assumptions, a universal form for the leading term, which we test on some treatable two-dimensional examples. We also derive a general upper bound on the entanglement, valid along the full time evolution. In two dimensions, the bound is computed analytically, while in higher dimensions it is evaluated at early and late time via the OPE. These CFT predictions are then compared with a doubly-holographic setup where the CFT is interpreted as a reservoir for the radiation produced on an end-of-the-world brane. After finding the gravitational dual of a boundary local quench, we compute the time evolution of the holographic entanglement entropy, whose late-time behavior is in perfect agreement with the CFT predictions. In the brane+bath picture, unitarity of the time evolution is preserved thanks to the formation of an island. The holographic results can be recovered explicitly from the island formula, in the limit where the tension of the brane is close to the maximal value.

KEYWORDS: Boundary conformal field theories, Page curve, Entanglement Entropy, Holography.

# 1 Introduction

This paper is devoted to the study of properties of the radiation injected into a conformal field theory (CFT) by a local modification of the vacuum. In particular, we consider a CFT with a boundary, and focus on the effect of an excitation of the boundary degrees of freedom. The ensuing burst of energy moves at the speed of light, and can be collected and studied at infinity. There are multiple reasons to be interested in this setup. For instance, boundaries can be engineered or be relevant in condensed matter experiments. Impurities are often modeled in the infrared as conformal boundary conditions [1, 2], and one can imagine preparing the impurity in an excited state, and measuring the excitations which propagate in the material once it relaxes to its ground state. More generally, the setup we consider has a certain degree of universality, and we expect observables of the kind discussed in this paper to arise in virtually any situation where real time dynamics of light degrees of freedom (the CFT) is considered in the presence of heavy degrees of freedom (the boundary). We will take the boundary of the system to preserve part of the conformal symmetry of the bulk, and create the excitation by acting with a local boundary operator on the vacuum. Similar local quenches have been considered before in homogeneous CFTs [3–5]. In this work, we prove various general results on the entanglement entropy and the energy density of the radiation, both in two and higher dimensions. In particular, we discuss a universal bound on entanglement, which turns out to be generically saturated at early and late times. To do so, we take advantage of an operator product expansion (OPE) whose convergence along the real time evolution of the system we prove. We also illustrate our results in a few examples, before moving on to strong coupling and entering the realm of holography.

Indeed, much of the recent interest in the real time dynamics of conformal field theories with boundaries comes from a theoretical motivation. In AdS/CFT [6], boundary CFTs (BCFTs) can be used as laboratories to study processes in quantum gravity which require degrees of freedom to be able to escape to infinity. Indeed, while CFTs per se provide a UV complete definition of quantum gravity in AdS, the required boundary conditions make energy conserved, and forbid radiation from trespassing the AdS boundary. Most notably, large black holes do not evaporate. In order to alleviate the problem, one can couple the AdS boundary to a reservoir [7], which can be taken to be a CFT without dynamical gravity. In this context, much of the recent progress in understanding the black hole information paradox [8] has happened [9–11]. The role of BCFT arises by taking the reservoir to be holographic as well, thus defining a *doubly-holographic* system. In one duality frame, the gravitating region is replaced by its boundary, and one is left with a BCFT. If, on the other hand, we replace the CFT degrees of freedom with their own gravity dual, we land on a description of the system which involves a higher dimensional asymptotically AdS space, ending on the union of a portion of the conformal boundary and the lower dimensional space where the evaporation process is taking place [11, 12]. The latter boundary can be modeled by an end-of-the-world (EoW) brane [13–16], where, contrary to the asymptotic boundary, the metric fluctuates. In view of the BCFT axioms, unitarity of the evaporation process is guaranteed, and in particular the entanglement entropy of the Hawking radiation must follow a Page curve [17]. In this context, the question is *how* to compute the Page curve. The mechanism by which unitarity is restored may well have to do with subtle properties of quantum gravity [18], but whatever the UV origin for this IR phenomenon is, it must change the rules by which entanglement is computed, if the paradox must be avoided.[1]

---

[1] The situation is somewhat analogous to the entropy of an ideal gas and the associated Gibbs paradox. In that case, the solution of the paradox lies in the indistinguishability of the particles, a deep quantum mechanical fact. On the other hand, the exact entropy can be computed just by modifying the rules of the game, and compensating for the overcounting.

It is remarkable that such new rules exist. If one wants to compute the entanglement entropy of a subregion $\Sigma$ of the bath, one is instructed to use island formula [9, 11, 19–21]:

$$S(\Sigma) = \min_I \left( \frac{A(\partial I)}{4G_N} + S_{\text{semi-classical}}(\Sigma \cup I) \right) . \tag{1.1}$$

Here, $I$ denotes the island: a second codimension one region located in the gravitating part of space-time. $A$ is the area of the boundary of the island, and $S_{\text{semi-classical}}$ is the entanglement entropy of the quantum fields, computed in the fixed semi-classical background.

The island formula has a two dimensional proof based on the Euclidean path integral [20, 22], but our understanding of the gravitational path integral is incomplete and at times puzzling [23], and it is useful to look for approaches which are controlled and manifestly UV complete. In the doubly-holographic setup, eq. (1.1) looks less surprising, because the appearance of the island can be understood from the properties of the Ryu-Takayanagi (RT) [24] surfaces which compute entanglement entropy in the higher dimensional duality frame [11]. In fact, the BCFT (or interface CFT) framework has proved valuable in understanding entanglement in quantum gravity beyond the black hole information paradox. In particular, it has become clear that islands are a generic property of the entanglement wedge of the radiation coming from a gravitating region, and are necessary to unitarize processes where the black holes may or may not be present [25–32].

Most of the quantitative analyses performed in this context have dealt with systems in the vacuum, or in thermal equilibrium. One of the purposes of the present work is to initiate the study of the entanglement of the radiation arising from a pure state, in a doubly-holographic setup. This involves, in the first place, studying the holographic dictionary that translates the local boundary quenches described above into geometric states in AdS. Once the dictionary is set up, we compute the holographic entanglement entropy of subsystems of the BCFT, and match the results to the dual picture, thus validating the dictionary itself. We also recover the result from the 'brane+bath' perspective, using the island formula (1.1). The match of the holographic result to the island formula is done in detail, in the case of a two-dimensional system, along the lines of what was recently done in [33, 34], and confirms the validity of eq. (1.1), in a quantitative way.

The paper is organized as follows. In section 2 we describe the setup in detail, discuss the observable central to this work and its available OPE channel. Section 3 is devoted to the study of entanglement entropy in the CFT, and to the derivation of a universal bound. The bound can be expressed in closed form for all times in two dimensions—eq. (3.37)—and it can be evaluated at early and late times in higher dimensions—eq. (3.52). In section (4), we exemplify the previous results via explicit computations using the replica trick. Section 5 is dedicated to heavy states in two dimensional holographic CFTs: there we study the gravity dual to the boundary quench, we compute the entanglement entropy via the RT prescription and reproduce it with the island formula, and discuss the energy density profile of the radiation. We conclude in section 6, with a few future directions.

**Note added:** *While this draft was being finalized, reference [35] appeared, which partially overlaps with our section 5, including in particular the study of the holographic dictionary for boundary local quenches.*

## 2 A pure excited state in a boundary CFT

Let us begin in two dimensions, for notational simplicity, although all the statements in this section immediately generalize to higher dimensions. Consider then a two dimensional conformal field theory (CFT), in Lorentzian signature, with a flat timelike boundary. We take the boundary conditions to preserve the $SO(1,2)$ group of conformal transformations which do not displace the boundary. Locally, the symmetry is enhanced to one copy of the Virasoro algebra. This setup is usually known as a boundary CFT (BCFT). We put the CFT in an excited state by acting with a local boundary operator at the origin of space at $t = 0$. More precisely, to ensure that the state is normalizable, we displace the insertion in Euclidean time:

$$|\mathcal{O}\rangle = \frac{1}{\sqrt{\langle \mathcal{O}(t = -i\epsilon)\mathcal{O}(t = i\epsilon)\rangle}} \, \mathcal{O}(x = 0, t = +i\epsilon) |0\rangle . \tag{2.1}$$

Notice that $\epsilon$ is a finite positive number. The presence of the boundary is understood in the definition of the vacuum $|0\rangle$, as will be in all correlation functions in this work. We will always take $\mathcal{O}$ to be a quasi-primary operator with scaling dimension $\Delta$. As it can be checked for instance measuring the expectation value of the stress tensor, this state is characterized by radiation incoming towards the boundary, being reflected and going back to infinity, see *e.g.* [36]. The setup is depicted in figure 1. Besides the expectation value of the stress tensor, we might be interested in measuring various other observables: for instance, the moments of the energy distribution, or the Rényi entropies associated to a subregion. Each of these observables contains information on the radiation. Since the state is not an eigenstate of the Hamiltonian, all of them will be time dependent. Nevertheless, it is important to notice that the state (2.1) is an eigenstate of a *different* conformal generator. Indeed,

$$\frac{1}{2}\left(\frac{K^0}{\epsilon} + \epsilon P^0\right)|\mathcal{O}\rangle = \Delta |\mathcal{O}\rangle . \tag{2.2}$$

If we were to use this charge as the generator of time translations, all correlation functions in this state would be time translational invariant. This is especially useful in the holographic context, since it implies that the dual geometry has a time-like Killing vector. We will exploit this fact in section 5.

In this work, we shall mostly be interested in the local properties of the radiation, as measured by the expectation value of local operators. Notice however that this set of observables includes the Rényi entropies associated to a semi-infinite interval, as we shall review below. It is therefore important to first distill the constraints imposed by conformal invariance and the operator product expansion (OPE) on the generic three-point function of a bulk and two boundary operators, which we do in the next subsection.

### 2.1 Analytic properties of the correlation function

Let us denote by $\phi$ a quasi-primary bulk operator, scalar for simplicity. Then the correlation function

$$\langle \mathcal{O}(\tau_1)\mathcal{O}(\tau_2)\phi(x, \tau)\rangle , \tag{2.3}$$

up to a trivial kinematical prefactor, is a function of a single cross ratio

$$\zeta = \frac{x^2 (\tau_1 - \tau_2)^2}{[x^2 + (\tau - \tau_1)^2][x^2 + (\tau - \tau_2)^2]} . \tag{2.4}$$

Here, $\tau$, $\tau_i \in \mathbb{R}$ are initially Euclidean directions. The correlator (2.3), for a particular ordering of operators in Lorentzian signature, computes the expectation value of $\phi$ in the state (2.1). Since we

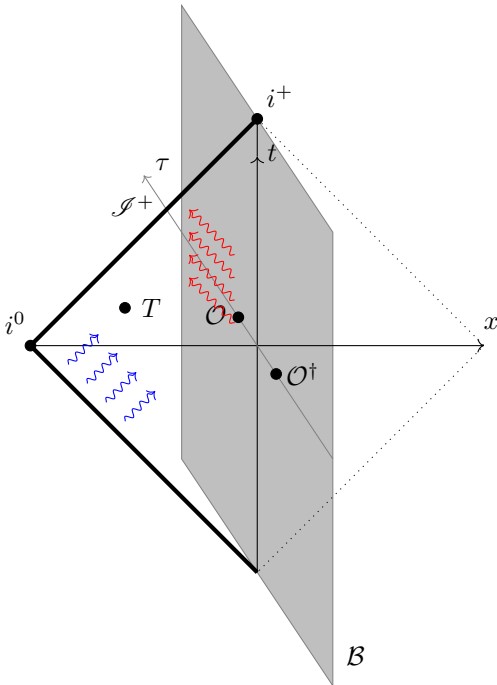

**Figure 1**: A sketch of the state considered in most of this paper. The Lorentzian dynamics involves energy reflecting off the boundary. It takes place in the $(x < 0, t)$ half-plane, which is here represented by its Penrose diagram. The Euclidean section is shaded, with the Euclidean time direction $\tau$ entering the page. The boundary operator $\mathcal{O}$ that creates and annihilates the state is displaced along $\tau$, and one insertion of a component of the stress tensor is shown for illustrative purposes.

care about the Lorentzian dynamics, we should check the analytic properties of the correlator along the Wick rotation contour [37]. These are most easily seen after the following change of variable:

$$\zeta = \frac{4r^2}{(1+r^2)^2} \; . \tag{2.5}$$

The new cross ratio $r$ has a simple geometric meaning, illustrated in figure 2. It is not hard to see that every Euclidean configuration for the correlator (2.3) can be brought to the one in figure 2 by a conformal transformation. In particular, notice that the range of $r$ can be restricted to $r \in [0, 1]$, because an inversion maps any larger value back to this range. Correspondingly, eq. (2.5) is invariant under $r \to 1/r$, and $\zeta \in [0, 1]$ in Euclidean signature.

The three-point function (2.3) can be expanded in powers of $r$, thanks to the boundary operator product expansion (OPE)

$$\phi(x, \tau) = \sum_i b_{\phi i} \, x^{\Delta_i - \Delta_\phi} \mathcal{O}_i(\tau) \; , \tag{2.6}$$

where $\mathcal{O}_i$ are boundary scaling operators with dimension $\Delta_i$, and $b_{\phi i}$ are real OPE coefficients in a unitary theory (if we choose a basis of real operators). In view of the usual radial quantization argument [38], the expansion of eq. (2.3) in powers of $r$ converges absolutely for $0 < |r| < 1$. In other words, $r$ is the $\rho$-coordinate for the three-point function under consideration [39]. As a consequence of

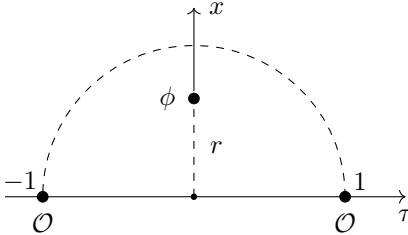

**Figure 2**: The cross ratio $\zeta$ takes the value in eq. (2.5) when the operators are placed as in the figure. The new cross ratio $r$ parametrizes the position of the bulk operator.

absolute convergence, the correlator can be analytically continued in the region $0 < |r| < 1$, where $r$ is promoted to a complex variable. Since the scaling dimensions appearing in eq. (2.6) are positive real numbers, $r = 0$ is generically a branch point, whose cut is convenient to place away from the $\Re r > 0$ half-plane. The open half-disk $|r| < 1$, $\Re r > 0$ is mapped to the complex plane of $\zeta$, excluding the two lines $\zeta \leq 0$ and $\zeta \geq 1$. We conclude that the correlator (2.3) is analytic there.

While the negative $\zeta$ axis is occupied by the cut dictated by the OPE (2.6), it is interesting to ask about the analytic properties on the $\zeta \geq 1$ line, which corresponds to $r = e^{i\theta}$, with $\theta \in [0, \pi/2]$ or $\theta \in [0, -\pi/2]$. For our purposes, it is sufficient to notice the following. Looking at figure 2, we see that nothing special happens at $r = 1$. More formally, the three-point function admits a Taylor expansion around this point. The expansion can again be interpreted in radial quantization, this time around $r = 1$, in terms of states created by bulk scaling operators—in fact, all descendants of $\phi$. It therefore converges absolutely in a disk of radius one, which means that the correlation function is analytic in this region, including at $r = 1$. Going back to $\zeta$, we conclude that there is no singularity in $\zeta = 1$.[2] This regularity property can sometimes be used to put constraints on the CFT data [40], and in a different context is at the heart of some versions of the bulk reconstruction program in AdS/CFT [41].

## 2.2 Expectation values in real time and the Operator Product Expansion

We are now ready to perform the Wick rotation and discuss the availability of an OPE in the configuration of interest, shown in figure 1, in that case with $\phi = T$. It turns out that this step is especially simple in our case. Consider measuring the expectation value of $\phi(x, t)$, for fixed $x$ at all times. Then, in eq. (2.4), $\tau_1 = -\tau_2 = \epsilon$, while $\tau = it$ is the only variable we will need to dial. Starting at $t = 0$, the configuration is Euclidean, so the correlator is evaluated on the Riemann sheet where the OPE converges, with $0 < \zeta(t = 0) < 1$. As time proceeds, the cross ratio is real and positive, and reaches the maximum value $\zeta = 1$ at $t = \sqrt{x^2 - \epsilon^2}$, while asymptotically

$$\zeta \underset{t \sim +\infty}{\sim} \frac{4\,\epsilon^2 x^2}{t^4} \ . \tag{2.7}$$

The function $\zeta(t)$ is shown in the left panel of figure 3. Since the correlator (2.3) is regular at $\zeta = 1$, the expectation value is bounded at all times, and more importantly the OPE always converges, except at most when $\zeta = 1$. The OPE (2.6) converges best around $\zeta = 0$, which is seen from eq. (2.7) to correspond to the late time limit. The fact that the late time asymptotics of the correlator is fixed by a convergent OPE is a valuable tool, and we will use it in the following to predict a largely universal behavior for the energy flux and the entanglement entropy of the radiation.

---

[2] The symmetry $r \to 1/r$ guarantees that no additional singularity is introduced in the change of coordinates, so that the correlator is single valued around $\zeta = 1$.

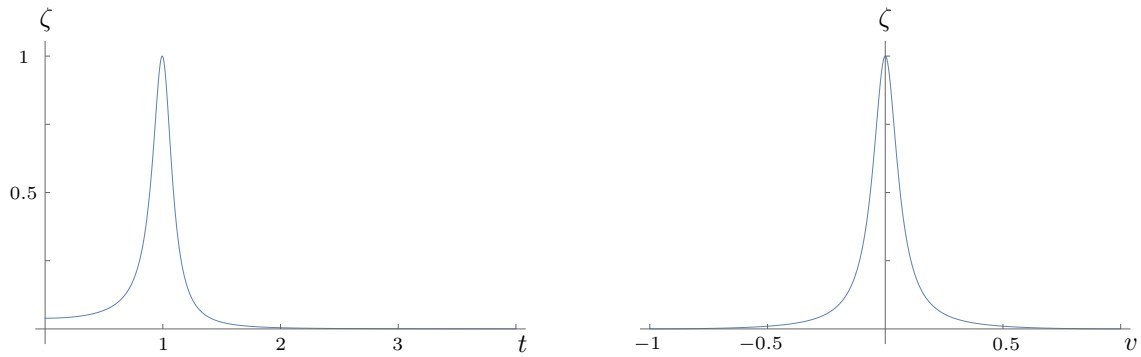

**Figure 3**: On the left, the cross ratio $\zeta$ as a function of time at fixed $x$ and $\epsilon$. On the right, the same cross-ratio as a function of $v$ on $\mathscr{I}^+$ at fixed $\epsilon$.

We will also be interested in measuring the expectation values of observables in the asymptotic region $\mathscr{I}^+$, as depicted in figure 4. In the next section, we shall illustrate the conceptual advantages of this configuration. In this case, it is convenient to consider the following compact coordinates

$$
\begin{aligned}
U &= \frac{2}{\pi} \arctan\left(\frac{x-t}{\ell}\right) , \\
V &= \frac{2}{\pi} \arctan\left(\frac{x+t}{\ell}\right) ,
\end{aligned}
\tag{2.8}
$$

which both take values between $-1$ and $1$. The cross-ratio takes the form

$$
\zeta = \frac{(\epsilon/\ell)^2 \left[\tan \frac{\pi U}{2} + \tan \frac{\pi V}{2}\right]^2}{\left[\tan^2 \frac{\pi U}{2} + (\epsilon/\ell)^2\right] \left[\tan^2 \frac{\pi V}{2} + (\epsilon/\ell)^2\right]} .
\tag{2.9}
$$

The arbitrary scale $\ell$ will be set to 1 in the following. Then, on $\mathscr{I}^+$, *i.e.* $U = -1$, one finds

$$
\zeta = \frac{\epsilon^2}{\tan^2\left(\frac{\pi V}{2}\right) + \epsilon^2} + \mathcal{O}(U+1) .
\tag{2.10}
$$

As shown in figures 3 and 4, the evolution of $\zeta$ is symmetric around $V \to -V$, and in particular both the early and late time[3] limits

$$
\zeta \underset{V\sim-1}{\sim} \frac{\pi^2 \epsilon^2}{4}(V+1)^2 , \qquad \zeta \underset{V\sim1}{\sim} \frac{\pi^2 \epsilon^2}{4}(V-1)^2 ,
\tag{2.11}
$$

are controlled by the same boundary OPE (2.6).

## 3 Universality in the entanglement of the radiation

We now focus on a specific expectation value, i.e. the one-point function of the twist operator in the excited state of the BCFT. This will allow us to extract universal features of the entropy of radiation at early and late time.

---

[3]We shall often use the inappropriate word "time" for the lightlike coordinate $V$.

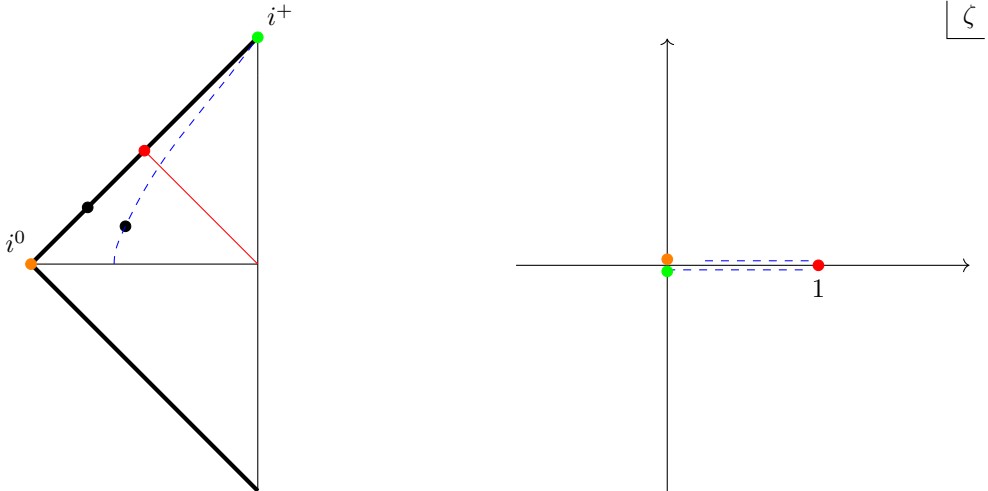

**Figure 4**: The time evolution of the observable (black dot) in compact coordinates (on the left) and in cross ratio space (on the right). As the operator moves along $\mathscr{I}^+$, the path in cross ratio space has a perfect $\mathbb{Z}_2$ symmetry. At $i^0$ and $i^+$, the value of the cross ratio is $\zeta = 0$, while on the light-cone emanating from the origin $\zeta = 1$. On the other hand, the time evolution at fixed $x$ starts at $t = 0$ with a generic value of $\zeta \in (0, 1)$. The path still reaches $\zeta = 1$ and goes towards $\zeta = 0$ at late time.

### 3.1 Replica computations and boundaries

We are interested in computing the entanglement entropy of the radiation produced by a boundary quench. Let us first consider the discussion in generic $d$ and then specialize to $d = 2$. The boundary is stretched along the time direction and $d-1$ spatial coordinates. Therefore, for a flat boundary, a time slice is just a half hyperplane of dimension $d - 1$. The codimension-two entangling surface divides the half hyperplane into two regions $A$ and $\bar{A}$ and the reduced density matrix is defined by integrating out the degrees of freedom of one of the two regions

$$\rho_A = \text{Tr}_{\bar{A}}(\rho) \tag{3.1}$$

We will pick the entangling surface to be a half sphere anchored on the boundary, see figure 7. In the two-dimensional case, it reduces to a point in the bulk. The physical reason for this choice is that the state (2.1) injects in the CFT a burst of spherically symmetric radiation, and so a spherical entangling surface is sufficient to probe its entanglement. This highly symmetric setup of course has technical advantages and it has been studied before for the vacuum case [42, 43].

The entanglement entropy is simply the von Neumann entropy associated to the reduced density matrix

$$S_{\text{EE}} = \text{Tr}(\rho_A \log \rho_A) \tag{3.2}$$

and it is useful to introduce the Rényi entropy

$$S_n = \frac{1}{1 - n} \log \text{Tr}(\rho_A^n) \tag{3.3}$$

a one-parameter generalization such that $\lim_{n \to 1} S_n = S_{\text{EE}}$. A common strategy to compute (3.3) is through the replica trick, where one computes $S_n$ using a path integral on a replicated manifold where

the different replicas are sewn together along codimension-one hypersurfaces ending on the entangling surface. For us, this manifold will be a $n$-fold half-hyperplane $\mathcal{H}_n$. The vacuum Rényi entropy is then given by

$$S_n = \frac{1}{1-n} \log \frac{Z[\mathcal{H}_n]}{Z[\mathcal{H}]^n} \ , \tag{3.4}$$

where $Z[\mathcal{M}]$ indicates the partition function calculated on the manifold $\mathcal{M}$. Equivalently, one can introduce a twist operator $\sigma_n$, i.e. an extended excitation localized on the entangling surface. $\sigma_n$ is best interpreted as a defect in the orbifolded tensor product $\text{CFT}^n/\mathbb{Z}_n$ [44–46]. The vacuum Rényi entropy in a BCFT is then computed by the expectation value of the twist operator in the presence of a boundary

$$S_n = \frac{1}{1-n} \log \langle \sigma_n \rangle \tag{3.5}$$

The generalization to an excited state $|\mathcal{O}\rangle$, such as the boundary quench introduced in section 2, is just

$$S_n^{\mathcal{O}} = \frac{1}{1-n} \log \langle \mathcal{O}^{\otimes n} | \sigma_n | \mathcal{O}^{\otimes n} \rangle \ , \tag{3.6}$$

where we introduced the notation $|\mathcal{O}^{\otimes n}\rangle$ to denote a state associated to a multiple-copy operator $\mathcal{O}^{\otimes n}$, i.e. a local operator inserted at the same point in each replica.

Both the quantities (3.6) and (3.5) are plagued by UV divergences associated to short-wavelength correlations near the entangling surface. A UV-finite quantity is provided by the excess of entropy

$$\Delta S_n = S_n^{\mathcal{O}} - S_n = \frac{1}{1-n} \log \frac{\langle \mathcal{O}^{\otimes n} | \sigma_n | \mathcal{O}^{\otimes n} \rangle}{\langle \sigma_n \rangle} \tag{3.7}$$

This is the observable of interest for our work. For integer $n$ in generic dimensions, the numerator of (3.7) is a 2-point correlator of multiple copy operators in the presence of two defects (the twist operator and the boundary). It is therefore highly non trivial to extract universal information on this quantity. Some simplification occurs in two dimensions and we now specialize to that case. On the other hand, in section 3.3, we will find a general bound for the entanglement of radiation in any dimension, i.e. for the quantity

$$\Delta S_{EE} = \lim_{n \to 1} \Delta S_n \ . \tag{3.8}$$

In two-dimensional CFTs, the twist operator is a local conformal primary of scaling dimension

$$\Delta_n = \frac{c}{12} \left( n - \frac{1}{n} \right). \tag{3.9}$$

and the numerator of (3.7) is effectively a three-point correlator in the presence of a boundary. Inserting a single twist operator in the bulk we are measuring the Rényi entropy of a spatial region extending from a point $x$, where the twist operator is located, to infinity. This setup is represented in the left panel of figure 5. Despite the physical process we have in mind involves an external injection of energy at a given time $t = 0$, the BCFT observable (3.7) we are considering is time-reversal invariant. Therefore, the outgoing radiation (represented in red in figure 5) is compensated by an ingoing radiation coming from past infinity (blue in figure 5). Since we are not interested in collecting the latter, at the end of the computation, we will move the twist operator towards the future lightcone $\mathscr{I}^+$ by taking the limit $U \to -1$ in the compact coordinates (2.8).

Our starting point is the Euclidean setup depicted in the upper left part of figure 9. The twist operator is inserted in a generic complex point $z_0$ with $\text{Re}(z_0) < 0$ with a branch cut extending from

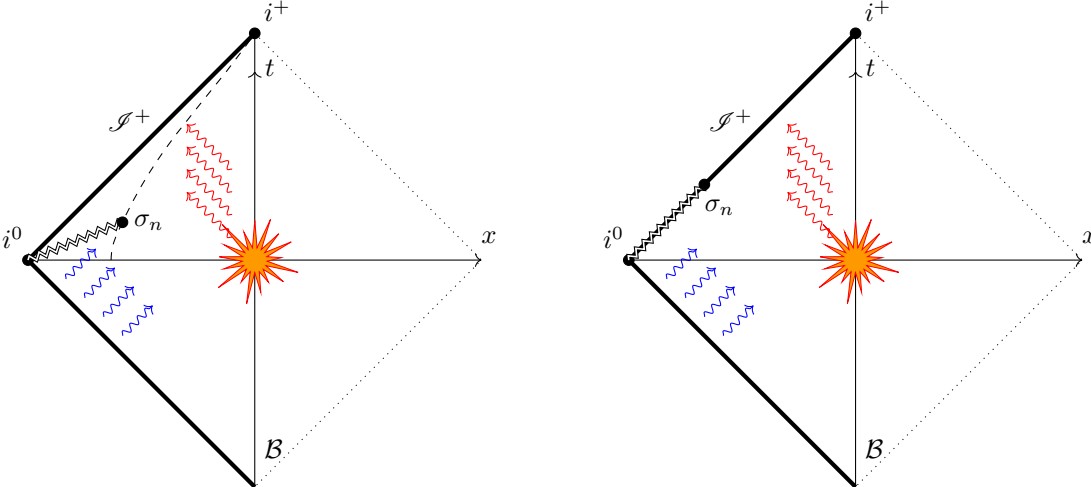

**Figure 5**: The Penrose diagram of the setup described in the main text. *Left panel.* The dashed line is the trajectory of the entangling surface (a point, in $2d$) which separates the spatial slice in two. *Right panel.* The twist operator in this case sits at $\mathscr{I}^+$.

there to the point at infinity. The multiple-copy operators are located at Euclidean time $\tau_1 = \epsilon$ and $\tau_2 = -\epsilon$. We need to compute

$$\Delta S_n = \frac{1}{1-n} \, \log \frac{\langle \sigma_n(z_0) \mathcal{O}^{\otimes n}(y_1) \mathcal{O}^{\otimes n}(y_2)\rangle}{\langle \sigma_n(z_0)\rangle \langle \mathcal{O}(y_1)\mathcal{O}(y_2)\rangle^n} \;. \tag{3.10}$$

where $z_0 = x + i\,\tau$, $y_1 = i\,\epsilon$ and $y_2 = -i\,\epsilon$ with $x \le 0$. Thanks to the normalization, the observable $\Delta S_n$ is conformal invariant if $\mathcal{O}$ is a Virasoro primary. Were $\mathcal{O}$ a quasi-primary, the quantity $\Delta S_n$ would be only $sl(2,\mathbb{C})$ invariant. In the following we will assume that $\mathcal{O}$ is a quasi-primary operator. As discussed in Section 3.2, the correlator (3.10) will depend on the single cross-ratio (2.4). We now consider the early- and late-time limit of $\Delta S_n$ and use the OPE to constrain those regimes.

## 3.2 The boundary OPE at early and late time

As we discussed in Section 2, the (early- and) late-time behaviour of a one-point function in an excited boundary state is controlled by the limit $\zeta \sim 0$, i.e. the only OPE channel that is available for this observable. Specifically, the lightest boundary operators appearing both in the OPE of two operators $\mathcal{O}$ and in the boundary OPE of the twist operator provides the leading contribution for $t \to 0$ and for $t \to \infty$. The lightest operator is clearly the identity, which leads to a disconnected contribution

$$\langle \sigma_n(z_0)\mathcal{O}^{\otimes n}(y_1)\mathcal{O}^{\otimes n}(y_2)\rangle \sim \langle \sigma_n(z_0)\rangle \langle \mathcal{O}(y_1)\mathcal{O}(y_2)\rangle^n \tag{3.11}$$

This cancels precisely the denominator in (3.10) yielding a vanishing contribution to $\Delta S_n$. In other words, as expected, the identity contribution equals the vacuum Rényi entropy. Therefore, the leading term for $\zeta \to 0$ is given by the lighest operator above the identity. Let us assume, for the moment, that this is the displacement operator. In a BCFT, the displacement can be easily introduced as the boundary limit of the orthogonal components of the bulk stress-tensor, whose defect OPE is always

non-singular. Its scaling dimension is then equal to the spacetime dimension $d$. For a two-dimensional BCFT, the displacement $D(\tau)$ for a boundary streched along the Euclidean time direction reads

$$D(\tau) = T(i\tau) + \bar{T}(-i\tau) \ , \tag{3.12}$$

which should be combined with Cardy's boundary conditions [47]

$$T(i\tau) = \bar{T}(-i\tau) \ . \tag{3.13}$$

The Zamolodchikov norm of the displacement operator is determined by the two-point function

$$\langle D(\tau)D(0) \rangle = \frac{C_D}{\tau^4} \ , \tag{3.14}$$

and in two dimensions it is fixed in terms of the central charge of the bulk CFT $C_D = 2c$.

The correlation function $\langle \mathcal{O}^{\otimes n} | \sigma_n | \mathcal{O}^{\otimes n} \rangle$ is defined on a $n$-fold cover of the spacetime, and the operators $\mathcal{O}^{\otimes n}$ creating the states are $n$-copy operators. Their OPE contains both single- and multiple-copy operators respecting the $\mathbb{Z}_n$ replica symmetry. The single-copy displacement operator is given by:

$$\mathcal{D}(\tau) = \sum_{m=0}^{n-1} \mathbb{1}^{\otimes m} \otimes D(\tau) \otimes \mathbb{1}^{\otimes(n-m-1)} \ . \tag{3.15}$$

and, for the moment, we assume it is the lightest operator which also appears in the defect OPE of the twist operator. Notice that the two-point function of $\mathcal{D}$ will be given by $C_{\mathcal{D}} = nC_D$. Under this assumption, we can expand the correlator relevant for (3.10) in the OPE limit $\zeta \to 0$ and we find that the leading contribution is given by

$$\frac{\langle \sigma_n(z_0) \mathcal{O}^{\otimes n}(\tau_1) \mathcal{O}^{\otimes n}(\tau_2) \rangle}{\langle \sigma_n(z_0) \rangle \langle \mathcal{O}(\tau_1) \mathcal{O}(\tau_2) \rangle^n} \sim \frac{c_{OO\mathcal{D}} b_{\sigma\mathcal{D}}}{C_{\mathcal{D}}} \zeta \tag{3.16}$$

where $c_{OO\mathcal{D}}$ is defined by the three-point function

$$\langle \mathcal{O}^{\otimes n}(\tau_1) \mathcal{O}^{\otimes n}(\tau_2) \mathcal{D}(\tau) \rangle = \frac{c_{OO\mathcal{D}}}{(\tau_1 - \tau_2)^{2n\Delta-2}(\tau_2 - \tau)^2(\tau_1 - \tau)^2} \tag{3.17}$$

whereas $b_{\sigma D}$ is the bulk-to-defect coupling of the twist operator and the displacement operator

$$\frac{\langle \mathcal{D}(\tau')\sigma_n(x+i\tau) \rangle}{\langle \sigma_n(x+i\tau) \rangle} = \frac{b_{\sigma\mathcal{D}} \, x^2}{(x^2 + (\tau - \tau')^2)^2} \tag{3.18}$$

We will determine the coefficients $c_{OO\mathcal{D}}$ and $b_{\sigma\mathcal{D}}$ using Ward identities, thus providing a universal contribution for the leading early- and late-time behaviour of the Rényi entropy (3.10).

To determine $c_{OO\mathcal{D}}$ it is important to remember that the stress tensor has a non-singular boundary OPE and its three-point function with two boundary operators is fixed by holomorphy to

$$\langle \mathcal{O}^{\otimes n}(\tau_1) \mathcal{O}^{\otimes n}(\tau_2) \mathcal{T}(z) \rangle = \frac{c_{OO\mathcal{T}}}{(\tau_1 - \tau_2)^{2n\Delta-2}(z - i\tau_2)^2(z - i\tau_1)^2} \tag{3.19}$$

and similarly for $\bar{\mathcal{T}}(\bar{z})$. Here the stress tensor $\mathcal{T}(z)$ (and $\bar{\mathcal{T}}(\bar{z})$) is understood as the single copy stress-tensor symmetrized over all the copies, analogously to what we did for the displacement operator in (3.15). Since the boundary OPE is non-singular, the Cardy gluing condition (3.13) imposes that $c_{OO\mathcal{T}} = c_{OO\bar{\mathcal{T}}}$. Using the definition of the displacement operator in (3.12) we conclude that $c_{OO\mathcal{D}} = 2c_{OO\mathcal{T}}$. Through the Ward identity

$$\int_0^\infty \frac{\mathrm{d}x}{2\pi} \langle \mathcal{O}^{\otimes n}(\tau_1) \mathcal{O}^{\otimes n}(\tau_2)(\mathcal{T}(x+i\tau) + \bar{\mathcal{T}}(x-i\tau)) \rangle = \partial_{\tau_1} \langle \mathcal{O}^{\otimes n}(\tau_1) \mathcal{O}^{\otimes n}(\tau_2) \rangle \ , \tag{3.20}$$

we conclude that

$$c_{OO\mathcal{T}} = n\Delta \ . \tag{3.21}$$

The two-point function of the twist operator with the single-copy displacement $\mathcal{D}(y)$ normalised by the vacuum expectation value of the twist operator is computed either from the contribution of the Schwarzian after the uniformising transformation [48, 49] or equivalently by using the Ward identity

$$\int \frac{d\tau'}{2\pi} \langle \mathcal{D}(\tau')\sigma_n(x)\rangle = \partial_x \langle \sigma_n(x)\rangle \tag{3.22}$$

From this we find

$$\langle \mathcal{D}(\tau')\sigma_n(x+i\tau)\rangle = \frac{-4\Delta_n x^2}{(x^2 + (\tau - \tau')^2)^2} \langle \sigma_n(x+i\tau)\rangle \ , \tag{3.23}$$

where $\Delta_n = \dfrac{c}{12}\left(n - \dfrac{1}{n}\right)$ is the scaling dimension of the twist operator. We then conclude that $b_{\sigma\mathcal{D}} = -4\Delta_n$. The above considerations completely fix the leading behaviour in the OPE limit $\zeta \to 0$ of (3.10):

$$\begin{aligned}
e^{(1-n)\Delta S_n} &= 1 - \frac{8n\Delta\,\Delta_n}{C_{\mathcal{D}}}\zeta + \mathcal{O}(\zeta^2) \\
&= 1 - \frac{\Delta(n^2-1)}{3n}\zeta + \mathcal{O}(\zeta^2) \ ,
\end{aligned} \tag{3.24}$$

which is universal and depends on the state only through its scaling dimension $\Delta$. Then the Rényi entropies take the form

$$\Delta S_n = \frac{n+1}{3n}\Delta\,\zeta + \mathcal{O}(\zeta^2) \ , \tag{3.25}$$

which can be trivially analytically continued to $n \to 1$ and gives the (early-) and late-time behaviour of the entanglement entropy:

$$\Delta S_{\mathrm{EE}} = \frac{2}{3}\Delta\,\zeta + \mathcal{O}(\zeta^2) \ . \tag{3.26}$$

An important assumption for our derivation is that the displacement is the lighest operator in the boundary OPE of $\sigma_n$. It is certainly true that the only operators appearing in the boundary OPE of $\sigma_n$ are multiple copy operators, except for the Virasoro identity module which includes also the displacement. This can be seen by performing the uniformizing transformation, under which the two-point function $\langle \mathcal{O}\sigma_n\rangle$ for a single copy boundary operator $\mathcal{O}$ is mapped to the one-point function $\langle \mathcal{O}\rangle$ which generically vanishes. The only exception is the identity module where the quasi-primaries in the OPE (e.g. the displacement) receive a Schwartzian contribution. Therefore, if we want to look for operators that are lighter than the displacement operator, we need to look for double copy operators. Suppose the lightest boundary operator in the BCFT $\mathcal{O}_L$ has dimension $\Delta_L$, then we can construct $n$ double copy operators of dimension $2\Delta_L$ of the form

$$[\mathcal{O}]_k^2 = \mathbb{1}^{\otimes m} \otimes \mathcal{O}_L(\tau) \otimes \mathbb{1}^{\otimes k} \otimes \mathcal{O}_L(\tau) \otimes \mathbb{1}^{\otimes(n-m-k-2)} + \mathbb{Z}_n\text{-symm} \tag{3.27}$$

From this analysis, we conclude that the displacement is certainly the lighest operator in the boundary OPE of $\sigma_n$ when

$$\Delta_L > 1 \ , \tag{3.28}$$

*i.e.*, in particular, when the boundary does not have relevant deformations. A further constraint arises if we restrict to holographic theories, i.e. when we take a large central charge $c$ and a sparse spectrum. In that case, all single trace operators are suppressed and the OPE of $\mathcal{O} \times \mathcal{O}$ for the single copy theory consists only of double trace operators. The lightest of such operators has dimension $2\Delta$ and since we need to consider a double-copy operator for it to have a non-vanishing OPE with $\sigma_n$, we have that the leading double trace contribution to the Rényi entropy comes from an operator of dimension $4\Delta$. We then conclude that, for holographic theories

| Range of $\Delta$ | Lightest operator | $\Delta S_n$ for $\zeta \to 0$ |
|---|---|---|
| $\Delta < 1/2$ | $[\mathcal{O}^2]_k^2$ | $G(n) \ \zeta^{2\Delta}$ |
| $\Delta > 1/2$ | $\mathcal{D}$ | $\frac{n+1}{3n}\Delta \ \zeta$ |

where the result for double trace operators is given in [5] and

$$G(n) = -\frac{n}{n-1} \sum_{k=1}^{n-1} \frac{1}{|\sin\left(\frac{\pi k}{n}\right)|^{4\Delta}} \tag{3.29}$$

In conclusion, for holographic theories the displacement always accounts for the leading term at late time if the boundary operator that generates the quench has dimension $\Delta > 1/2$. For non-holographic theories, instead the requirement is that the OPE $\mathcal{O} \times \mathcal{O}$ does not contain an operator with $\Delta_L < 1$.

## 3.3 A bound on the entanglement of the radiation

Based on our construction, we can expect that the excess of entanglement entropy (3.8) vanishes at late time. A less trivial question is whether we can find a universal bound on its (early- and) late-time fall-off. Here we successfully address this question analytically using the relative entropy bound [50]. We will show that the late-time behaviour of the entanglement entropy is bounded by a function which only depends on the scaling dimension of the primary operator creating the state $\Delta$ and we show that this bound is optimal by discussing examples where it is saturated.

The Bekenstein bound [51] is a universal bound on the entropy of a region in flat space:

$$S \leq k \, E \, L \ , \tag{3.30}$$

where $E$ is the total energy contained in the region, $L$ is a characteristic length of the system and $k$ is a numerical constant of order one. Casini [50] interpreted the left-hand side of the bound as a renormalised Von Neumann entropy $S(\rho_A) - S(\rho_A^0)$ of some region $A$, where $\rho_A$ and $\rho_A^0$ are the reduced density matrices of the excited state and the vacuum respectively. This is precisely the excess of entanglement entropy introduced in (3.8). We stress again that this difference is finite and well defined because the divergences do not depend on the state, but rather on geometric properties of $A$ [52]. The right-hand side, instead, is interpreted as the renormalised expectation value of the vacuum modular Hamiltonian $K^0$, defined by

$$\rho_A^0 = \frac{e^{-K^0}}{\text{Tr} \, e^{-K^0}} \ , \tag{3.31}$$

up to a constant shift $K^0 \to K^0 + \text{constant}$. Then the bound is re-formulated as

$$S(\rho_A) - S(\rho_A^0) \leq \text{Tr} K^0 \rho_A - \text{Tr} K^0 \rho_A^0 \ , \tag{3.32}$$

which is manifestly finite and well defined (in particular it is invariant under a constant shift of the Modular Hamiltonian), when the reduced density matrices are properly normalized $\text{Tr}\rho_A = \text{Tr}\rho_A^0 = 1$.

The bound (3.32) is a simple consequence of the positivity of the relative entropy between the local density matrices of the excited and the vacuum state reduced to $A$:

$$0 \leq S(\rho_A | \rho_A^0) = \mathrm{Tr}\rho_A \log \rho_A - \mathrm{Tr}\rho_A \log \rho_A^0 \ . \tag{3.33}$$

The relative entropy between two density matrices $S(\rho|\rho')$ is a measure of the statistical distance between the two states and it is always positive. Clearly, this has important consequences for our setup as it immediately provides an upper bound for our observable (3.8). Let us show how to compute the r.h.s. of (3.33).

### 3.3.1 Relative Entropy Bound in $d = 2$

For our two-dimensional setup, the region $A$ is the interval extending from $-\infty$ to the position $-x$ of the twist operator. The associated modular Hamiltonian for the vacuum state admits a local expression[4] [53, 54]:

$$K^0(x) = 2\pi \int_{-x}^{0} \frac{x^2 - X^2}{2x} T^{00}(X) \mathrm{d}X \ . \tag{3.34}$$

Let us consider the excited state produced by the boundary quench described in Section 2. Since it is a pure state, its density matrix is simply

$$\rho = |\mathcal{O}\rangle\langle\mathcal{O}|, \tag{3.35}$$

where we used the definition (2.1). We want to compute the expectation value of the modular Hamiltonian on the reduced density matrix $\rho_A$, but thanks to the simple form (3.34) we actually need to integrate the expectation value of $\mathrm{Tr}(T^{00}\rho_A)$ on the region $A$. By definition of reduced density matrix, for a local operator $\mathcal{O}_A$ in the region $A$ we have $\mathrm{Tr}(\mathcal{O}_A\rho_A) = \mathrm{Tr}(\mathcal{O}_A\rho)$, therefore we can simply compute $\mathrm{Tr}(\rho K^0)$. The expectation value of the modular Hamiltonian is then completely fixed by the three-point function of the operator $\mathcal{O}$ with the stress-tensor:

$$\begin{aligned}
\mathrm{Tr}(\rho K^0) &= 2\pi \int_{-x}^{0} \frac{x^2 - X^2}{2\,x} \langle \mathcal{O}|T^{00}(t, X)|\mathcal{O}\rangle \, \mathrm{d}X \\
&= 2\Delta + \mathrm{i}\,\Delta \frac{x^2 - t^2 - \epsilon^2}{2x\epsilon} \log \frac{t^2 - (x + i\epsilon)^2}{t^2 - (x - i\epsilon)^2} \\
&= 2\Delta - 2\Delta \sqrt{\frac{1 - \zeta}{\zeta}} \arcsin \sqrt{\zeta} \ .
\end{aligned} \tag{3.36}$$

The second term on the r.h.s. of eq. (3.32) involves the expectation value of the stress tensor in the vacuum, and therefore vanishes. Hence we directly get the bound

$$\Delta S_{EE} \leq 2\Delta - 2\Delta \sqrt{\frac{1 - \zeta}{\zeta}} \arcsin \sqrt{\zeta} \simeq \frac{2}{3}\Delta\,\zeta + \mathcal{O}(\zeta^2) \ . \tag{3.37}$$

Notice that the bound only depends on the state through the scaling dimension $\Delta$ and spread $\epsilon$. Furthermore, at late-time it saturates precisely the result (3.26) obtained considering only the displacement operator in the boundary OPE of $\sigma_n$. This implies an important result: *After subtracting the contribution of the displacement operator, the lighest operator in the boundary OPE of $\sigma_n$ must*

---

[4]In general, it is not possible to find a local expression for the modular Hamiltonian associated to a given region $A$. This happens only for very symmetric circumstances, like planar of spherical entangling surfaces.

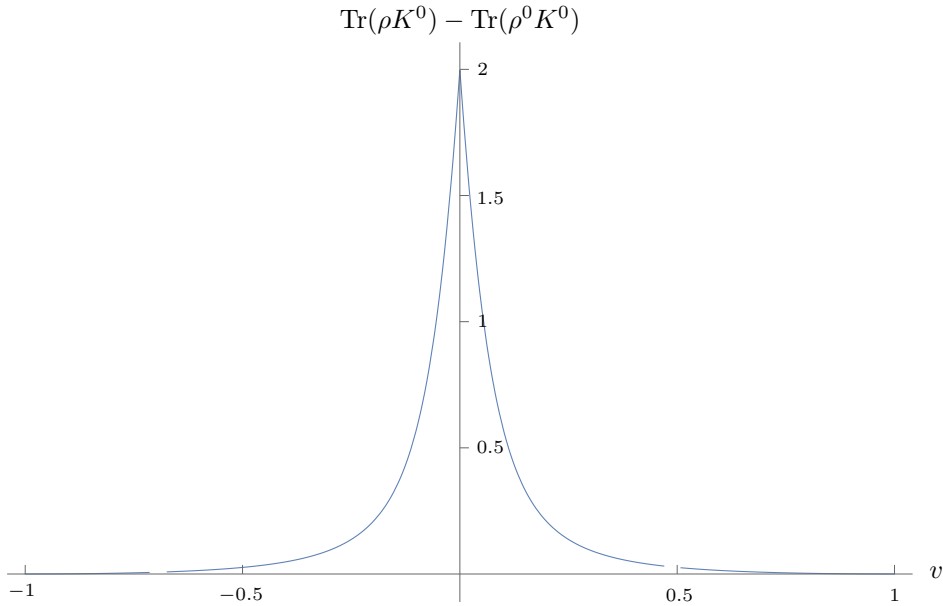

**Figure 6**: The bound on $\mathscr{I}^+$ as a function of $v \in [-1, 1]$ at fixed $\epsilon$.

*give a negative contribution to* $\Delta S_{EE}$. In particular, this is true also for the cases described at the end of section 3.2 when the OPE contains operators that are lighter than the displacement. An explicit example is the double trace contribution (3.29) which gives

$$\Delta S_{\text{EE}} = G(1)\zeta^{2\Delta} + \mathcal{O}(\zeta^{2\Delta+1}) \, , \tag{3.38}$$

with [5]

$$G(1) = -\frac{\Gamma\left(\frac{3}{2}\right)\Gamma\left(2\Delta+1\right)}{\Gamma\left(2\Delta+\frac{3}{2}\right)} < 0 \, . \tag{3.39}$$

The time evolution of the bound in the variable $V$ introduced in (2.8) is presented in figure 6.

We emphasize that $\Delta S_{\text{EE}}$ is the excess of entanglement entropy with respect to the vacuum so it is not necessarily positive. From the examples in Section 4 we will see that the existence of a lower bound on $\Delta S_{\text{EE}}$ is very unlikely.

### 3.3.2 Relative Entropy Bound in $d \geq 3$

The two-dimensional setup can be easily generalised to higher dimensions, where the twist operator is a codimension-2 conformal defect [45]. In $d \geq 3$ the radiation can also propagate on the boundary, so the natural generalisation of the $d = 2$ case is considering the twist-operator as a half-sphere $S^{d-2}$, which intersects the boundary on a $S^{d-3}$, as shown in figure 7.

The excess of Rényi entropy (3.7) for the density matrix reduced to the region $A$, enclosed by the twist operator $\Sigma_n(t)$ is then given by

$$\Delta S_n = \frac{1}{1-n} \log \frac{\langle \Sigma_n(t) \, \mathcal{O}^{\otimes n}(-i\epsilon, 0) \, \mathcal{O}^{\otimes n}(i\epsilon, 0)\rangle}{\langle \Sigma_n(t)\rangle_{\mathcal{B}} \langle \mathcal{O}(-i\epsilon, 0) \, \mathcal{O}(i\epsilon, 0)\rangle^n} \, , \tag{3.40}$$

The early- and late-time behaviours are controlled by the OPE channel where the two boundary operators, or equivalently the twist operator, are expanded in terms of local boundary operators (for

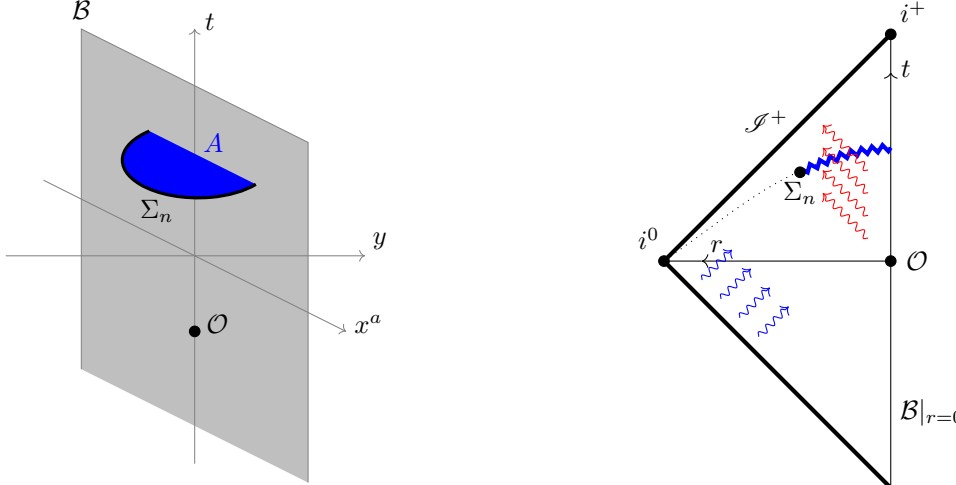

**Figure 7**: The generalisation of the set-up to higher dimensions: the twist operator $\Sigma_n$ (in black) is an half-sphere anchored to the boundary and centred on the (space) origin, enclosing the spatial region $A$ (in blue). On the right figure, the points in the Penrose diagram correspond to half-spheres anchored to the boundary.

the twist operator, this it the limit where the hemisphere shrinks to zero on top of the boundary). The convergence of the OPE can be made manifest in $\rho$-coordinates. For simplicity, we only treat in detail the three dimensional case. They basic observation is that, since conformal transformations map spheres into spheres, the entangling surface can be fully specified by its intersection with the boundary. We start from the Euclidean configuration of our set-up with the two boundary operators at $x_{1,2} = (\pm\epsilon, 0, 0)$ and the two points given by the intersection of the entangling surface with the boundary in $x_{3,4} = (\tau, \pm R, 0)$.[5] Recall that the first coordinate denotes Euclidean time. This configuration can be mapped via conformal transformations [38, 39] to one which makes the convergence of the OPE manifest, shown in figure 8: the position of the boundary is unchanged, while the two operators are mapped to the points $x'_{12} = (\pm 1, \vec{0})$. The twist operator is still a $(d-2)$-dimensional hemi-circle and it is parametrized by $x'_{34} = (\pm r \cos\theta, \pm r \sin\theta, 0)$, where

$$\rho = re^{i\theta} \tag{3.41}$$

is fixed by

$$z = \frac{4\rho}{(1+\rho)^2} \tag{3.42}$$

with

$$u = \frac{x_{12}^2 x_{34}^2}{x_{13}^2 x_{24}^2} = z\bar{z} \ , \qquad v = \frac{x_{23}^2 x_{14}^2}{x_{13}^2 x_{24}^2} = (1-z)(1-\bar{z}) \ . \tag{3.43}$$

In particular, in our configuration we have $x_{23}^2 x_{14}^2 = x_{13}^2 x_{24}^2$, which implies $\theta = \frac{\pi}{2}$. The whole dependence on $\tau, \epsilon$ and $R$ is now encoded in the radius $r$ of the hemisphere in $\rho$-coordinates and the OPE converges if $r < 1$. Therefore, to consider the Lorentzian time evolution, $\tau = it$, at fixed $R$ and $\epsilon$ we

---

[5]The generalisation to higher dimensions requires considering a spherical surface on the boundary, whose properties under conformal transformations are captured by the $\rho$ coordinates defined in [55].

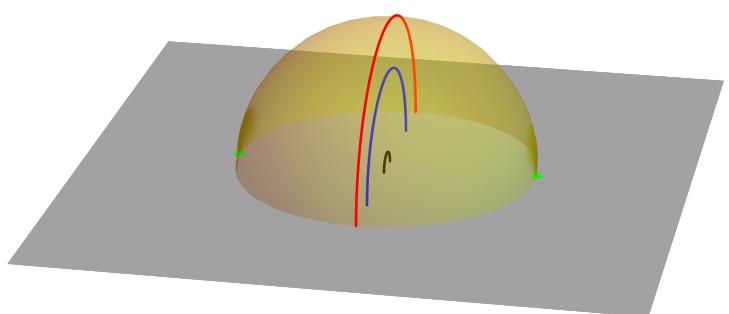

**Figure 8**: The configuration in $\rho$-coordinates of for our setup in $d \geq 2$ is the following: the boundary is at $y = 0$ (in grey) and we have two boundary operators creating the states at $(-1, 0, 0)$ and $(1, 0, 0)$ (the green dots), the twist operator parametrised by $(0, -r\,\Omega^{\mathfrak{a}}\cos\alpha, r\sin\alpha)$ for $\alpha \in [0, \pi]$, where $\Omega^{\mathfrak{a}}$ parametrises a unit $(d-3)$-dimensional sphere $(\Omega^{\mathfrak{a}}\Omega^{\mathfrak{b}}\delta_{\mathfrak{a}\mathfrak{b}} = 1)$. In the figure we show three different configurations of the twist operator: • at early time $t = 0$ (in black), • at $t = \sqrt{R^2 - \epsilon^2}$ (in red), • and at late time $t > \sqrt{R^2 - \epsilon^2}$ (in blue).

can simply analytically continue the function $u$ and analyze the radius of the hemisphere at various instants in time (notice that $u$ is still real for $\tau = it$). The cross-ratio is defined in the range $0 < u \leq 4$ and it reaches its maximum for $t = \sqrt{R^2 - \epsilon^2}$. The relation between $r$ and $u$ is

$$r = \frac{2 - \sqrt{4 - u}}{\sqrt{u}} \ , \tag{3.44}$$

then we find $0 < r \leq 1$ and the OPE is convergent both at early time and late time (see figure 8) and at $t = \sqrt{R^2 - \epsilon^2}$ the entangling surface is on the unit sphere (red curve in figure 8). To reach this conclusion, it is important that the path in cross ratio space is the same for the Lorentzian time evolution and for the Euclidean evolution depicted in figure 8, so that the correlator at late time is equal to the Euclidean counterpart at $r < 1$, which has a convergent OPE.

This OPE can be used to study the early- and late-time limits of the entanglement entropy, analogously to what we did in two dimensions. Instead, let us consider the bound (3.32) in this setup. Luckily, the modular Hamiltonian corresponding to the vacuum density matrix reduced on an half-sphere in $d$-dimensions still takes a local form [54]

$$K_A^0(R, t) = 2\pi \int_A \mathrm{d}^{d-1}x\, \frac{R^2 - \vec{x}^2}{2R}\, T^{00}(t, \vec{x}) \ , \tag{3.45}$$

where $A$ is a space-like half-sphere, *i.e.* $\vec{x}^2 \leq R^2$ and $y \leq 0$[6].

As in the two-dimensional case then, the computation reduces to the integral of the expectation value of the stress tensor in the excited boundary state

$$\mathrm{Tr}(K_A^0 \rho_A) = 2\pi \int_A \mathrm{d}^{d-1}x\, \frac{R^2 - \vec{x}^2}{2R}\, \langle \mathcal{O}|T^{00}(t, \vec{x})|\mathcal{O}\rangle \ , \tag{3.46}$$

---

[6]In the Euclidean, we split a generic vector as $x^\mu = (x^0, \vec{x}) = (x^0, x^{\mathfrak{a}}, y) = (x^a, y)$, where $\mu = 0, \ldots, d-1$ and $y$ is the direction orthogonal to the boundary $\mathcal{B}$.

Contrary to 2d, however, the three-point function with the boundary $\langle \mathcal{O}|T^{00}(t, \vec{x})|\mathcal{O}\rangle$ is not fixed (in 2d holomorphicity for the stress tensor allows to fix the correlator completely) and depends on the conformal cross ratio $\zeta$. We can still use the OPE to bound the late-time behaviour of the excess of entanglement. The component $T^{00}$ is parallel to the boundary, so we should first discuss the spectrum of operators that are contained in the boundary OPE of $T^{ab}$. The displacement certainly appears, but in principle one could also have a boundary spin-two operator $W^{ab}$ of dimension[7] $d - 1 < \Delta_W < d$ which gives a singular contribution, such that $T^{ab}(y) = b_{TW}y^{\Delta_W - d}W^{ab} + b_{TD}D\delta^{ab} + \dots$. Here we assume that this operator is absent, i.e. we take the displacement as the lightest operator in the boundary OPE of $T^{ab}$. Under this assumption, we can write

$$\frac{\langle T^{00}(t, \vec{x})\, \mathcal{O}(-i\epsilon, 0)\, \mathcal{O}(i\epsilon, 0)\rangle}{\langle \mathcal{O}(-i\epsilon, 0)\, \mathcal{O}(i\epsilon, 0)\rangle} \simeq \frac{c_{\mathcal{O}\mathcal{O}D}}{C_D}\, (2\epsilon)^d\, \langle T^{00}(t, \vec{x})D(0, 0)\rangle + \cdots , \tag{3.47}$$

where the dots stand for the contribution of heavier operators and $c_{\mathcal{O}\mathcal{O}D}$ is the coefficient of the boundary 3-point function $\langle \mathcal{O}\mathcal{O}D\rangle$.

The bulk-boundary two-point function $\langle T^{ab}(x^a, y)D(0, 0)\rangle$ is completely fixed by conformal symmetry [56] and in Euclidean signature it is given by

$$\langle T^{ab}(x^a, y)D(0, 0)\rangle = \frac{C_D}{d - 1} \frac{4d\, y^2 x_a x_b - \delta_{ab}(x^2 + y^2)^2}{(x^2 + y^2)^{d+2}} , \tag{3.48}$$

Continuing to Lorentzian signature, the correlator relevant for (3.47) is given by

$$\langle T^{00}(t, x^{\mathfrak{a}}, y)D(0, 0)\rangle = \frac{C_D}{d - 1} \frac{4d\, y^2 t^2 + (t^2 - x^2 - y^2)^2}{(t^2 - x^2 - y^2)^{d+2}} , \tag{3.49}$$

where in this equation we used $x^2 = x^{\mathfrak{a}}x_{\mathfrak{a}}$. We perform the integral in (3.46) by using coordinates $x^{\mathfrak{a}} = r\Omega^{\mathfrak{a}}\cos\theta$ and $y = r\sin\theta$ with $\theta \in [-\pi/2, 0]$

$$\langle K_A(t)D(0, 0)\rangle = \frac{2\pi C_D\, \Omega_{d-3}}{d - 1} \int_0^R dr \int_{-\frac{\pi}{2}}^0 d\theta\, r^{d-2}\cos^{d-3}\theta\, \frac{R^2 - r^2}{2R} \cdot \frac{4d\, t^2\, r^2\sin^2\theta + (t^2 - r^2)^2}{(t^2 - r^2)^{d+2}}$$

$$= C_D\, \frac{\pi^{\frac{d+1}{2}} R^d}{2\,(d-1)\,\Gamma\left[\frac{d+3}{2}\right](t^2 - R^2)^d} \simeq C_D\, \frac{\pi^{\frac{d+1}{2}} R^d}{2\,(d-1)\,\Gamma\left[\frac{d+3}{2}\right](t^2)^d} , \tag{3.50}$$

where $\Omega_{d-3} = 2\pi^{\frac{d}{2}-1}/\Gamma\left[\frac{d}{2} - 1\right]$ is the volume of $S^{d-3}$ and in the last line we have considered the OPE limit $R/t \to 0$. Combining (3.46) and (3.47) we get

$$\text{Tr}(K_A^0\rho_A) = c_{\mathcal{O}\mathcal{O}D} \cdot \frac{2^{d-1}\pi^{\frac{d+1}{2}}}{(d - 1)\,\Gamma\left[\frac{d+3}{2}\right]} \cdot \left(\frac{\epsilon\, R}{t^2}\right)^d + \dots , \tag{3.51}$$

where the dots refer to subleading terms. Since the stress tensor does not get an expectation value in the presence of a boundary, we have $\text{Tr}(K_A^0\rho_A^0) = 0$ and we can just insert (3.51) into (3.32) to obtain a sharp bound for the excess of entanglement entropy at late time:

$$\Delta S_{\text{EE}} \lesssim c_{\mathcal{O}\mathcal{O}D} \cdot \frac{2^{d-1}\pi^{\frac{d+1}{2}}}{(d - 1)\,\Gamma\left[\frac{d+3}{2}\right]} \cdot \left(\frac{\epsilon\, R}{t^2}\right)^d , \tag{3.52}$$

---

[7]The lower bound on the $\Delta_W$ is imposed by unitarity, while the upper bound is the request that this tensor is lighter than the displacement operator

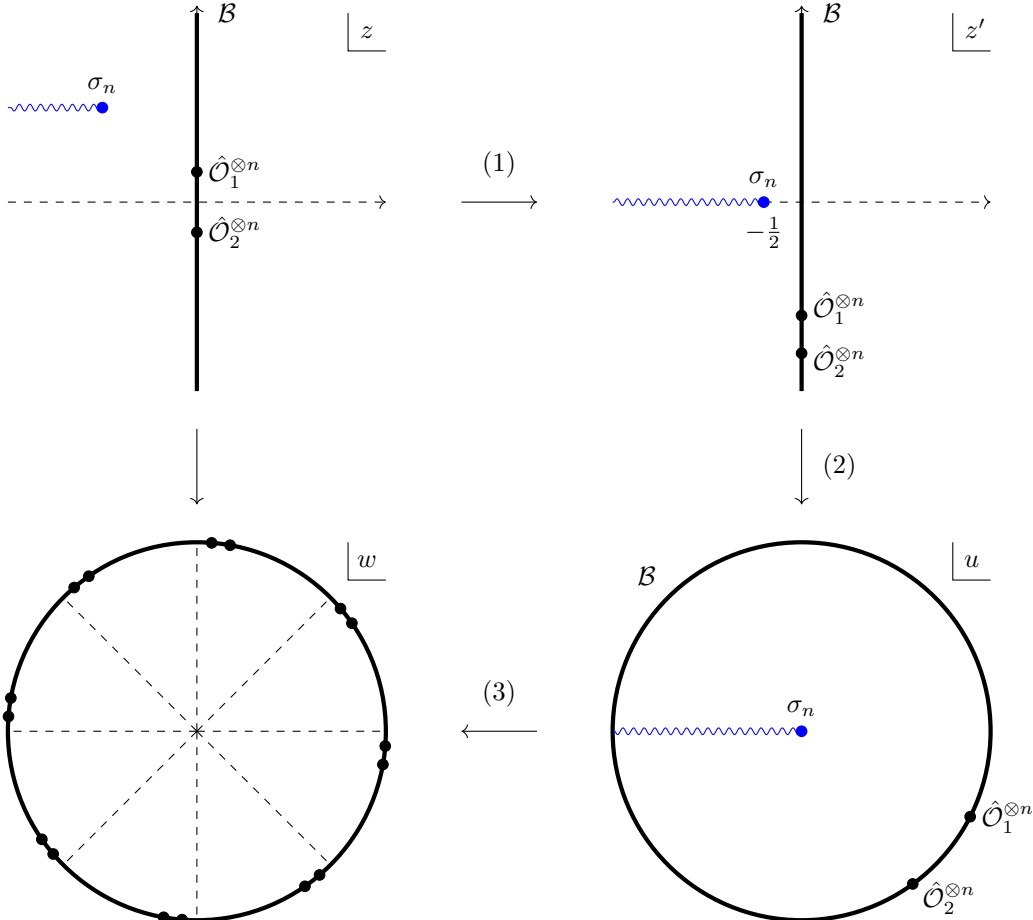

**Figure 9**: The series of conformal transformations mapping our main observable (3.10) to a $2n$-correlator on the unit disk. The explicit expressions are given in the main text.

which exactly matches the result (3.36) in two dimensions with $c_{\mathcal{O}\mathcal{O}D} = \Delta/\pi$. Notice that the quantity in brackets is the limit for $t \to \infty$ of the higher dimensional generalization of the cross-ratio (2.4), where $x$ is replaced by $R$. Following the argument in section 2, $\Delta S_{\mathrm{EE}}$ is a function of the cross-ratio and the early-time can be obtained from the first term on the second line of equation (3.51), taking the $t \to 0$ limit.

## 4 Light excited states

In this section we are going to study the time evolution of the entanglement entropy in two simple setups. First, we consider the Ising Model and the state created by the fermion operator on the boundary. Then, we consider the state created by the stress tensor on the boundary in a generic CFT. In the former case we are able to compute analytically the entanglement entropy at all times, while in the latter we present the result of the firsts Renyi entropies and the early- and late-time behaviour of the $S_n$ for any $n$. In both cases, the bound is perfectly saturated.

## 4.1 General strategy

In two dimensions, we can perform a series of conformal transformations to map the observable (3.10) to a $2n$-point correlator on the unit disk. The transformation (1) in Figure 9

$$z' = \frac{i\,\tau - z}{2x} \ , \tag{4.1}$$

maps the twist operators $\sigma_n$ to the position $z' = -\frac{1}{2}$. We then map the half-plane $\mathrm{Re}[z'] \leq 0$ to a disc of unit radius $\mathcal{D}$ ($|u| \leq 1$)

$$u = -\frac{z' + \frac{1}{2}}{z' - \frac{1}{2}} \ , \qquad z' = \frac{u-1}{2(u+1)} \ , \tag{4.2}$$

The boundary is now the circle $|u| = 1$ and the twist operator is in the origin.

So far, we only performed global conformal transformations and all the Jacobian factors in (3.10) are compensated by the normalisation in the denominator. Therefore, the excess of entropy in the $u$-plane is a three-point function of two multiple copy operators and a twist operator evaluated on the $n$-fold unit disc

$$\Delta S_n = \frac{1}{1-n} \ \log \frac{\langle \sigma_n(0)\mathcal{O}^{\otimes n}(u_1)\mathcal{O}^{\otimes n}(u_2)\rangle}{\langle \sigma_n(0)\rangle\langle\mathcal{O}(u_1)\mathcal{O}(u_2)\rangle^n} \ . \tag{4.3}$$

where the dependence on $x$, $\tau$ and $\epsilon$ is now contained in the two phases $u_1 = \frac{x+i(\tau-\epsilon)}{x-i(\tau-\epsilon)}$ and $u_2 = \frac{x+i(\tau+\epsilon)}{x-i(\tau+\epsilon)}$. Using the Virasoro extension of the conformal group in two dimensions, we can also perform a holomorphic transformation to map the $n$-fold disc to the unfolded disc $\mathcal{D}$

$$w = \sqrt[n]{u} \ . \tag{4.4}$$

For convenience, we only perform this transformation in the numerator of (4.3), i.e. we use the identity

$$\frac{\langle \sigma_n(0)\mathcal{O}^{\otimes n}(u_1)\mathcal{O}^{\otimes n}(u_2)\rangle}{\langle \sigma_n(0)\rangle} = \prod_{j=1}^n \left(\frac{du}{dw}\right)_{w=w_{1,j}}^{-\Delta} \left(\frac{du}{dw}\right)_{w=w_{2,j}}^{-\Delta} \langle \bigotimes_{l=1}^n \mathcal{O}(w_{1,l})\mathcal{O}(w_{2,l})\rangle \tag{4.5}$$

where $\Delta$ is the conformal dimension of the boundary operator $\mathcal{O}$ and

$$w_{k,l} = \sqrt[n]{u_k}e^{\frac{2i\pi l}{n}} \ . \tag{4.6}$$

This transformation maps the three-point function with two multiple copy operators to a $(2n)$-point function on the boundary of $\mathcal{D}$, giving

$$e^{(1-n)\Delta S_n} = \prod_{j=1}^n \left(n^2 w_{1,j}^{n-1} w_{2,j}^{n-1}\right)^{-\Delta} \frac{\langle\bigotimes_{l=1}^n \mathcal{O}(w_{1,l})\mathcal{O}(w_{2,l})\rangle}{\langle\mathcal{O}(u_1)\mathcal{O}(u_2)\rangle^n} \ , \tag{4.7}$$

As in [5], we can organize the right hand side in a *universal* contribution which do not depend on the details of the theory and a *dynamical* contribution:

$$\Delta S_n = \Delta S_n^{\mathrm{univ}} + \Delta S_n^{\mathrm{dyn}} \ , \tag{4.8}$$

where

$$e^{(1-n)\Delta S_n^{\mathrm{univ}}} = \prod_{\substack{j=1,\ldots,n \\ i=1,2}} \left(n^2 w_{1,j}^{n-1} w_{2,j}^{n-1}\right)^{-\Delta} \frac{\prod_{i=1}^n \langle\mathcal{O}(w_{1,i})\mathcal{O}(w_{2,i})\rangle}{\langle\mathcal{O}(u_1)\mathcal{O}(u_2)\rangle^n} \ , \tag{4.9}$$

$$e^{(1-n)\Delta S_n^{\mathrm{dyn}}} = \frac{\langle\bigotimes_{i=1}^n \mathcal{O}(w_{1,i})\mathcal{O}(w_{2,i})\rangle}{\prod_{i=1}^n \langle\mathcal{O}(w_{1,i})\mathcal{O}(w_{2,i})\rangle} \ . \tag{4.10}$$

The universal contribution can be evaluated explicitly for any $n$ and the analytic continuation to $n \to 1$ is trivial:

$$e^{(1-n)\Delta S_n^{\text{univ}}} = \left[ \frac{(u_1 \, u_2)^{1-n}}{n^{2n}} \cdot \left( \frac{u_1 - u_2}{u_1^{1/n} - u_2^{1/n}} \right)^2 \right]^{\Delta} , $$

$$\Delta S_{\text{EE}}^{\text{univ}} = \Delta \left( 2 - \frac{u_1 + u_2}{u_1 - u_2} \log \frac{u_1}{u_2} \right) . \tag{4.11}$$

On the other hand, the dynamical contribution is theory dependent and we can only extract universal information in the OPE limit. It seems we did not gain much from this decomposition. However, comparing the result (4.11) with the bound (3.36) we notice that they are perfectly matching. This means that the universal part already saturates the relative entropy upper bound and any contribution that is added must be negative

$$\Delta S_{\text{EE}}^{\text{dyn}} \leq 0 . \tag{4.12}$$

## 4.2 The Ising model

In this section we consider the theory of a free fermion with a boundary, which corresponds to the two-dimensional Ising model. We take a boundary state created by the fermionic operator defined as the limiting value of the holomorphic and anti-holomorphic fermion operator defined in the bulk. In particular, the two-point functions of the fermionic operators in the BCFT are

$$\langle \psi(z_1)\psi(z_2) \rangle = \frac{1}{z_1 - z_2} , \qquad \langle \bar{\psi}(\bar{z}_1)\bar{\psi}(\bar{z}_2) \rangle = \frac{1}{\bar{z}_1 - \bar{z}_2} , \qquad \langle \psi(z_1)\bar{\psi}(\bar{z}_2) \rangle = \frac{1}{z_1 - \bar{z}_2} , \tag{4.13}$$

and $\psi(i\tau) = \bar{\psi}(-i\tau)$. Then, we can use formula (4.7) to compute the $(2n)$-point function of the fermion operators on the boundary of the disk

$$\langle \psi(u_1)\psi(u_2) \rangle = \frac{1}{u_1 - u_2} , \tag{4.14}$$

This yields the Renyi entropies for any integer $n$ as a polynomial of degree $\lfloor \frac{n}{2} \rfloor$ in the cross-ratio

$$\zeta = \frac{1}{4} \left( 2 - \frac{u_1}{u_2} - \frac{u_2}{u_1} \right) . \tag{4.15}$$

The result of this computation is

$$e^{(1-n)\,\Delta S_n} = \prod_{i=1}^{\lfloor \frac{n}{2} \rfloor} \left[ 1 - \frac{(n - 2i + 1)^2}{n^2} \zeta \right] , \tag{4.16}$$

This expression has been found from analytic computation for $n = 2, \ldots, 6$ and checked numerically for higher $n$. Taking the logarithm of this expression, expanding the function around $\zeta = 0$ and performing the resulting sum over $i$, we find

$$\Delta S_n = \frac{1}{1-n} \sum_{k=1}^{\infty} \frac{1}{k} \left( \frac{4\zeta}{n^2} \right)^k \zeta(-2k, \frac{n+1}{2}) , \tag{4.17}$$

where $\zeta(s,q)$ is the *Hurwitz zeta* function. For integer values and $n \geq 2$ this series is convergent. The analytic continuation to $n \sim 1$ is more subtle because the series does not converge any more, for

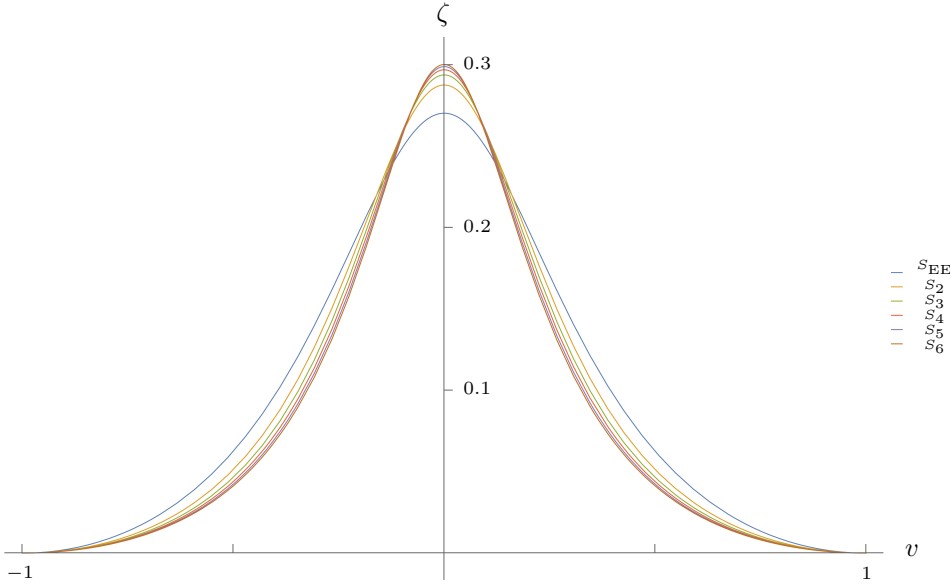

**Figure 10**: The time evolution of the entanglement entropy and the first Renyi entropies in compact coordinates at $\mathscr{I}^+$. In this graph, we have chosen $\epsilon = 1$ in (2.10).

non-integer values of $n$. Nevertheless, we can formally interpret our result as an asymptotic expansion in $\zeta$, which is analytic in $n$. The series we obtain for the entanglement entropy, for $n = 1$, coincides exactly with the asymptotic expansion of the *digamma* function $\psi(z) \equiv \frac{\Gamma'(z)}{\Gamma(z)}$ around $z = \infty$

$$
\begin{aligned}
\Delta S_{\text{EE}} &= \frac{1}{2} \sum_{k=1}^{\infty} \frac{(4\zeta)^k}{k} B_{2k} \\
&= \log\left(\frac{1}{2\sqrt{\zeta}}\right) - \psi\left(\frac{1}{2\sqrt{\zeta}}\right) - \sqrt{\zeta} \ ,
\end{aligned}
\tag{4.18}
$$

where $B_n$ are the Bernoulli numbers. In Figure 10, we show the time evolution at $\mathscr{I}^+$ for $S_{\text{EE}}$ and the Renyi entropies for $n = 2, \ldots, 6$. Although our derivation was not rigorous, we notice the the position of the curve is consistent with the extrapolation of the result for higher $n$. Furthermore, expanding around $\zeta \to 0$

$$
\Delta S_{\text{EE}} = \frac{\zeta}{3} - \frac{2\zeta^2}{15} + \mathcal{O}(\zeta^3) \ ,
\tag{4.19}
$$

we find perfect agreement with the behaviour predicted in (3.26) for $\Delta = 1/2$ confirming that no operator lighter than the displacement is exchanged in the OPE. Furthermore, consistently with the bound (3.37), the first subleading correction is negative.

## 4.3 The stress tensor state in a generic CFT

We now consider a generic two-dimensional CFT and the boundary state created by acting with the stress tensor at the origin. The bulk two-point functions of the holomorphic and anti-holomorphic

components of the stress tensor can be computed using the so-called doubling trick [47]:

$$\langle T(z_1)T(z_2)\rangle = \frac{c}{2(z_1-z_2)^4} \;, \qquad \langle \bar{T}(\bar{z}_1)\bar{T}(\bar{z}_2)\rangle = \frac{c}{2(\bar{z}_1-\bar{z}_2)^4} \;, \qquad \langle T(z_1)\bar{T}(\bar{z}_2)\rangle = \frac{c}{2(z_1-\bar{z}_2)^4} \;, \tag{4.20}$$

where $c$ is the *central charge*. The boundary value of the stress tensor $T(i\tau) = \bar{T}(-i\tau)$ correspond to the displacement operator on the conformal boundary, already mentioned in Section 3.2.

The Renyi entropies can be computed again using the replica trick on the disk

$$e^{(1-n)\Delta S_n} = \frac{\langle \sigma(0)\,T^{\otimes n}(u_1)T^{\otimes n}(u_2)\rangle}{\langle \sigma(0)\rangle\langle T(u_1)T(u_2)\rangle^n} \;, \tag{4.21}$$

but now we have to keep in mind that the stress tensor is not a Virasoro primary, then equation (4.7) does not apply. Indeed, under the uniformising map the stress tensor transforms as

$$T(u) = \left(\frac{\mathrm{d}u}{\mathrm{d}w}\right)^{-2}\left[T'(w) - \frac{c}{12}\{u,w\}\right] \;, \tag{4.22}$$

where the Schwarzian derivative is

$$\{u,w\} = \frac{1-n^2}{2\,w^2} \;. \tag{4.23}$$

Then equation (4.21) becomes a sum over correlation functions of $m \leq 2n$ stress tensors multiplied the proper Jacobian factors and normalisation, times $2n - m$ Schwarzian derivatives. The generic $n$-point correlation function of stress-energy tensor can be computed using the recursion relation [57]

$$\begin{aligned}
\langle T(z_1)\cdots T(z_n)\rangle &= \sum_{i=2}^{n}\left[\frac{2}{(z_1-z_i)^2} + \frac{1}{z_1-z_i}\partial_{z_i}\right]\langle T(z_2)\cdots T(z_n)\rangle \\
&+ \sum_{i=2}^{n}\frac{c}{2(z_1-z_i)^4}\,\langle T(z_2)\cdots T(z_{i-1})T(z_{i+1})\cdots T(z_n)\rangle \;,
\end{aligned} \tag{4.24}$$

which gives us the analytic form of the Renyi entropies. In Figure 11, we show the time evolution of $S_n$ for $n = 2,3,4$, in CFTs with different central charges. We can notice that the $c \to \infty$ limit does not commute with either $n \to 1$ or $\zeta \to 0$. Indeed, the contribution of the Schwarzian, which is leading at large $c$, vanishes in the limit $n \to 1$ and is the less singular contribution for $\zeta \to 0$.

A closed formula for the Rényi entropies of the state created by the stress tensor would be rather convoluted. A more modest task is the computation of the early- and late-time behaviour for any $n$, which turns out to be

$$\Delta S_n \simeq \frac{1}{1-n}\log\left(\frac{(u_1 u_2)^{2-2n}}{n^{4n}}\left(\frac{u_1-u_2}{u_1^{1/n}-u_2^{1/n}}\right)^{4n}\right) = \frac{2(n+1)}{3n}\zeta + \mathcal{O}(\zeta^2) \;, \tag{4.25}$$

which perfectly agrees with (3.25) for $\Delta = 2$. This is expected since the OPE of two displacement operators does not contain any operator that is lighter than the displacement itself and it is a strong consistency check of our results. The limit $n \to 1$ is simply

$$\Delta S_{EE} = \frac{4}{3}\zeta + \mathcal{O}(\zeta^2) \;. \tag{4.26}$$

in agreement with (3.26).

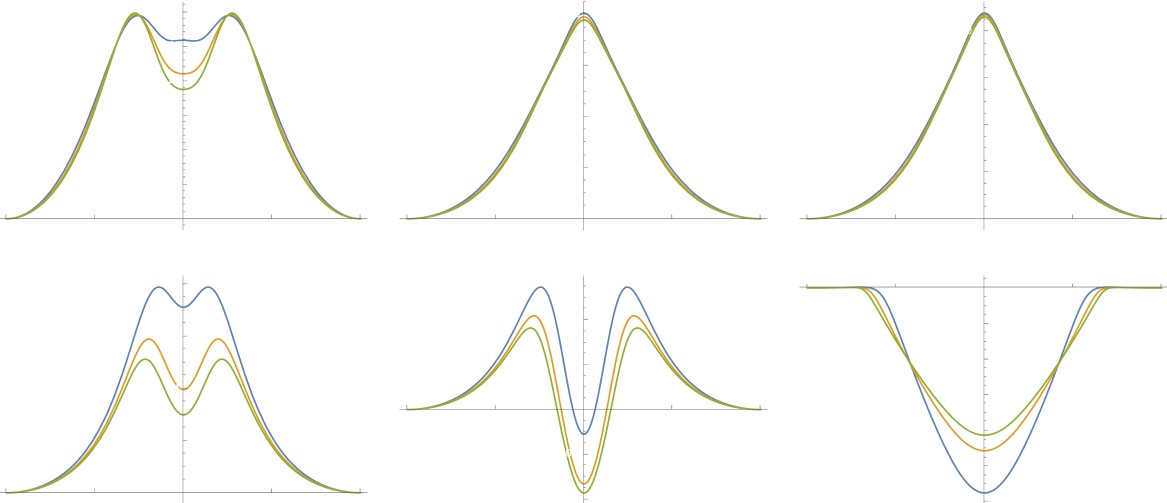

**Figure 11**: The time evolution of the first Renyi entropies at $\mathscr{I}^+$ for the state created by the stress tensor on the boundary with $\epsilon = 1$, in CFTs with different central charges: in order $c = \frac{1}{2}$, $c = \frac{11}{12}$, $c = 1$, $c = 10$, $c = 20$, $c = 10^4$.

## 5   Heavy excited states and holography

This section is dedicated to the study of a heavy excited state in a holographic boundary CFT. We take the boundary operator $\mathcal{O}$ in eq. (2.1) with scaling dimension $\Delta$ of the same order as the central charge $c$, which is a large number. We will work in the semi-classical approximation, disregarding $1/c$ corrections. The CFT also has a large gap above the Virasoro identity module, so that its dual is approximately described by pure gravity in a locally AdS$_3$ background [58]. We also need to pick a conformal boundary condition, and for this we turn to the bottom up model described in [15], where the asymptotically AdS bulk is cut off by an end-of-the-world (EoW) brane, anchored at the location of the boundary in the BCFT. The action on the gravity side is

$$S = -\frac{1}{16\pi G}\int_{\text{Bulk}} \mathrm{d}x^3 \sqrt{g}\left(R + \frac{2}{L^2}\right) + \lambda \int_{\text{Brane}} \mathrm{d}s^2 \sqrt{\hat{g}} + \frac{1}{8\pi G}\int_{\partial(\text{Bulk})} \mathrm{d}s^2 \sqrt{\hat{g}}K + \ldots \qquad (5.1)$$

where $\hat{g}$ is the induced metric on the brane and the dots stand for the counterterms and Hayward term at the intersection of the brane with the AdS boundary—see *e.g.* appendix A of [59]. In order to arrive at the gravity dual of the state in eq. (2.1), we shall first review the dictionary in the absence of the EoW brane, in subsection (5.1). We shall include the brane in subsection 5.2, where we formulate the holographic dictionary between heavy boundary operators and AdS geometries. Then, we move on to measuring the flux of energy and the entanglement entropy of the radiation: matching the results to the CFT computations, we find justification of the proposed dictionary. Finally, in subsection 5.4, we re-interpret our computations in the doubly holographic framework, and we show that the entanglement entropy computed holographically implies the validity of the island formula.

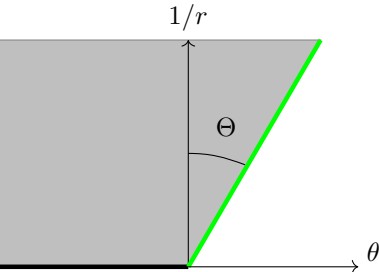

**Figure 12**: A time slice of the region close to the AdS boundary with the EoW brane ending on the conformal defect.

## 5.1 Heavy operators in AdS/CFT

Let us first consider the state created by a heavy local operator, as in eq. (2.1), but in a ordinary homogeneous CFT. The vacuum state of the CFT is dual to empty AdS$_3$ with radius $L$ obeying [60]

$$c = \frac{3L}{2G_N} \ . \tag{5.2}$$

Without the boundary, the holographic dual of a local excitation is given by a massive particle falling in Poincaré AdS [4]. The construction of the back-reacted metric in this set-up was given in [61]. The key is exploiting eq. (2.2). This equation says that the state (2.1) is an eigenstate of the conformal Hamiltonian [62], which is nothing but the generator of time translation on the Lorentzian cylinder. Of course, different values of $\epsilon$ are related by a dilatation, so the parameter reflects the ambiguity of the change of coordinates between Minkowski space and the cylinder.[8] The corresponding Killing vector of empty Poincaré AdS is

$$t \to t/\alpha \ , \quad x \to x/\alpha \ , \quad z \to z/\alpha \ , \qquad \alpha > 0 \ , \tag{5.3}$$

where as usual

$$ds^2_{\text{Poincaré}} = L^2 \left( \frac{-dt^2 + dz^2 + dx^2}{z^2} \right) \ . \tag{5.4}$$

Composing the diffeomorphism (5.3) with the change of coordinates from global AdS to the Poincaré patch [63], we obtain the one-parameter family of transformations which on the conformal boundary connect Minkowski space and the Lorentzian cylinder:

$$\begin{aligned}
\sqrt{L^2 + r^2} \cos t_g &= \frac{\alpha L^2 + (z^2 + x^2 - t^2)/\alpha}{2z} \ , \\
\sqrt{L^2 + r^2} \sin t_g &= \frac{Lt}{z} \ , \\
r \cos \theta &= \frac{Lx}{z} \ , \\
r \sin \theta &= \frac{-\alpha L^2 + (z^2 + x^2 - t^2)/\alpha}{2z} \ .
\end{aligned} \tag{5.5}$$

If we now fix $\alpha$ so that the generator in (2.2) is mapped to the translation of $t_g$, we obtain a frame where the geometry produced by $|\mathcal{O}\rangle$ is static. This is easily seen to be

$$\alpha = \frac{\epsilon}{L} \ . \tag{5.6}$$

---

[8]The same ambiguity showed up in the choice of coordinates on the Penrose diagram, and was parametrized by $\ell$ in eq. (2.8).

If the operator $\mathcal{O}$ is a single trace, as we shall assume, the geometry is specifically the result of the backreaction of a particle sitting at the center of AdS. We are led to the following metric:

$$\mathrm{d}s^2 = -(r^2 - qL^2)\mathrm{d}t_g^2 + \frac{L^2\mathrm{d}r^2}{r^2 - qL^2} + r^2\mathrm{d}\theta^2 \ , \tag{5.7}$$

where $\theta$ is periodic with period $2\pi$. Here, we have

$$q = \frac{M}{L^2} - 1 \ , \tag{5.8}$$

where $L$ is the radius of the asymptotic AdS$_3$ and $M$ is proportional to the mass $m$ of the particle at rest at $r = 0$ [64]. We will distinguish three cases:

1. $q = -1$, which corresponds to $M = 0$, *i.e.* pure AdS$_3$,

2. $-1 < q < 0$, corresponding in AdS$_3$ to a conical singularity with deficit angle,

3. $q > 0$, which is a black hole solution with a Schwarzschild radius $r_H = \sqrt{q}\,L$.

The precise correspondence between $q$ and the scaling dimension $\Delta$ is obtained by measuring the stress tensor via the Fefferman-Graham expansion of the metric [65], as per the rules of holographic renormalization [66]. The result is

$$q = \frac{12\Delta}{c} - 1 \tag{5.9}$$

The description of the state in the Poincaré patch is finally obtained via eqs. (5.5,5.6). Notice in particular that this diffeomorphism maps the trajectory $r = 0$ in global coordinates to

$$x = 0 \ , \qquad z^2 - t^2 = \epsilon^2 \ , \tag{5.10}$$

in the Poincaré patch. This confirms that the dual to eq. (2.1), in the absence of a boundary, is a particle that enters the Poincaré horizon, falls to a minimum distance from the boundary fixed by $\epsilon$, and accelerates away towards the future horizon.

## 5.2 The holographic dual of a heavy boundary operator

We can combine the analysis of the previous subsection with the AdS/BCFT described by the action (5.1) to find the holographic dual of a boundary quench. The equations of motion obtained from the action (5.1) include the Israel-Lancsoz conditions [67, 68]

$$K_{ab} - K\hat{g}_{ab} = \lambda\,\hat{g}_{ab} \ , \tag{5.11}$$

or equivalently

$$K_{ab} = -\lambda\,\hat{g}_{ab} \ . \tag{5.12}$$

Following in the footsteps of the previous discussion, we start by looking for a geometry dual to an eigenstate of the cylinder Hamiltonian. More precisely, we place the CFT on an interval, so that the spacetime geometry is a Lorentzian strip. This renders our setup similar to the interface considered in [59], and we shall be able to follow in part the analysis performed there. The solution is given by the metric (5.7), restricted to the portion of space between the conformal boundary and the EoW brane. The location of the latter is obtained by solving eq. (5.12).

Since we look for a static solution, the embedding of the brane is specified by a single function $\theta(r)$, which we want to find. There is an obvious $\mathbb{Z}_2$ symmetry, and the parametrization $\theta(r)$ only

describes half of the brane—see figure 13. Furthermore, in the static case we can parametrize the induced metric as

$$d\hat{s}^2 = -f(r)dt_g^2 + g(r)dr^2 \ , \tag{5.13}$$

where the components are obtained by pulling back the metric (5.7) on the brane

$$f(r) = r^2 - qL^2 \ , $$
$$g(r) = \frac{L^2}{f(r)} + r^2 \left(\frac{d\theta}{dr}\right)^2 \ . \tag{5.14}$$

The matching condition (5.12) reduces to a single first-order differential equation

$$\frac{r^2}{L} \cdot \frac{d\theta}{dr} = -\lambda\sqrt{f(r)g(r)} \ , \tag{5.15}$$

where we took our boundary $CFT_2$ to live in $\theta \leq 0$. It is useful to first expand near the conformal boundary $(r \to \infty)$, where the parameter $M$ can be neglected:

$$ds^2 \simeq -r^2 dt_g^2 + \frac{l^2 dr^2}{r^2} + r^2 d\theta^2 \ , $$
$$f(r) \simeq r^2 \ , $$
$$g(r) \simeq \frac{L^2}{r^2} + r^2 \left(\frac{d\theta}{dr}\right)^2 \ . \tag{5.16}$$

We define the angle $\Theta$ in the $(\theta, \frac{1}{r})$ plane between the normal to the conformal boundary and the EoW brane as

$$\theta(r) \simeq \frac{L}{r} \tan\Theta \ , \tag{5.17}$$

as shown in figure 12.

Then, the matching conditions fix the $\Theta$ in terms of the tension of the brane $\lambda$ and the AdS radius $L$:

$$\sin\Theta = L\lambda \ . \tag{5.18}$$

The geometry on the brane, asymptotically close to the conformal boundary, is $AdS_2$, with radius

$$L_{\text{Brane}} = L\sec\Theta. \tag{5.19}$$

Notice that the tension $\lambda$ is bounded as a consequence of eq. (5.18),

$$-\frac{1}{L} \leq \lambda \leq \frac{1}{L} \ , \tag{5.20}$$

and that in the limit of maximal tension, $L_{\text{Brane}}$ diverges, and $\Theta \to \pi/2$, so that the brane flattens out and reconstructs the missing piece of the conformal boundary.

We can go back to the full matching condition (5.15) and extract

$$\frac{d\theta}{dr} = -\frac{L\tan\Theta}{r\sqrt{r^2 + qL^2\tan^2\Theta}} \ , \tag{5.21}$$

and from this the $(r,r)$ component of the induced metric:

$$g(r) = \frac{r^2 L^2}{(r^2 - qL^2)\left[(1 - L^2\lambda^2)r^2 + qL^4\lambda^2\right]} \ . \tag{5.22}$$

At this point, we need to differentiate the analysis depending on the value of $q$. Furthermore, we first focus on a brane with positive tension $\lambda \geq 0$. We comment on the opposite case in subsection 5.2.3.

### 5.2.1 $-1 \leq q \leq 0$: the conical singularity

When $q$ is negative in eq. (5.7), it is convenient to allow for $\theta$ to have an arbitrary period $2\pi\gamma$, with $\gamma > 0$. To see why, notice that eq. (5.21) is first order. Therefore, once we choose that the brane intersects the conformal boundary at $\theta = 0$, we cannot fix the second intersection point at will. Nevertheless, we would like to insist that the position of the boundaries of the strip in the dual CFT is at $\theta = 0$ and $\theta = -\pi$, so that we can apply the same coordinate transformation as in the homogeneous case—eq. (5.5)—to obtain the causal diamond shown in figure 1 on half of the Poincaré patch. The parameter $\gamma$ will allow us to do exactly that. In fact, while this change of coordinates works at the conformal boundary, it is not invertible on a full constant global time slice. This does not pose a problem when computing quantities in the CFT.

The solution of eq. (5.21) obeying the boundary condition (5.17) for $\Theta > 0$, reads

$$r(\theta) = \sqrt{-q}\,\frac{L\tan\Theta}{\sin(\sqrt{-q}\,\theta)} \ . \tag{5.23}$$

The brane meets the boundary at $\theta = 0$, and then again at $\theta = \pi/\sqrt{-q}$. The closest approach to the conical singularity happens at $\theta = \pi/2\sqrt{-q}$, and the conical singularity $r = 0$ is included in the spacetime, see figure 13. Since in our conventions the CFT lives *on the left* of the point $\theta = 0$, we can use the periodicity in $\theta$ to select the width of the strip:

$$\theta = \pi/\sqrt{-q} \sim \pi/\sqrt{-q} - 2\pi\gamma = -\pi \qquad \Longrightarrow \qquad \gamma = \frac{1}{2}\left(1 + \frac{1}{\sqrt{-q}}\right) \ . \tag{5.24}$$

Notice in particular that when $q = -1$, $\gamma = 1$ and we recover the vacuum state. Vice versa, when $q \to 0$, the periodicity diverges. If $q = 0$ the brane passes through the point $r = 0$, and there is no sense in which $\theta$ is periodic. As we will see in the next subsection, the geometry at this value transitions to a BTZ black brane.

We can now match this solution to the CFT, for instance by measuring the expectation value of the stress tensor on the Poincaré section. In view of our choice of coordinates, the result is identical to the vacuum in a CFT without a boundary [4]

$$\langle T_{\pm\pm}\rangle = \frac{M\epsilon^2}{8\pi G_N R((t\pm x)^2 + \epsilon^2)} \ . \tag{5.25}$$

This must be matched to the result from boundary CFT:

$$\langle \mathcal{O}|T(z)|\mathcal{O}\rangle = \frac{4\Delta\,\epsilon^2}{z^2 + \epsilon^2} \ , \qquad \langle \mathcal{O}|\bar{T}(z)|\mathcal{O}\rangle = \frac{4\Delta\,\epsilon^2}{\bar{z}^2 + \epsilon^2} \ , \tag{5.26}$$

so that (notice the factor 2 with respect to the homogeneous case eq. (5.9))

$$\frac{M}{4G_N R} = 4\Delta \ , \qquad q = \frac{24\Delta}{c} - 1 \ . \tag{5.27}$$

While our choice of periodicity was designed to obtain the simple relation (5.27) between $q$ and $\Delta$, it is worth emphasizing that the relation between $q$ and the conical deficit around $r = 0$ is different from the homogeneous case. Specifically, the opening angle of the cone is $2\pi(1 - \delta)$ with $\delta = (1 - \sqrt{-q})/2$, *i.e.* the deficit it is half of the case without a boundary.

When computing observables, as for instance the Rényi entropies in the next section, it might be convenient to pick a different frame, which makes it simple to exploit results already available for the

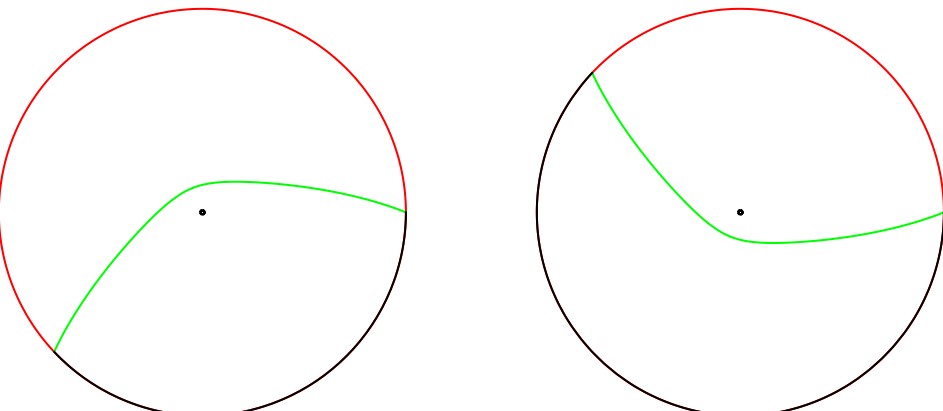

**Figure 13**: The EoW brane (in green) configurations with a conical singularity in the bulk, respectively for $\Theta > 0$ and $\Theta < 0$. The pictures represent a constant $t_g$ slice of global AdS, with the angular coordinate $\theta$ defined with period $2\pi$ and the radial coordinate $\tilde{r} = \frac{2}{\pi}\arctan r$. The CFT is defined in the black portion of the boundary. When the tension is positive, the conical singularity is in the bulk, while a brane with negative tension hides the conical singularity and the configuration is equivalent to the vacuum.

case without a boundary. Since the new frame offers an equivalent perspective for the computation of this subsection, let us spend a few words about it. The two dimensional infinite strip has a conformal Killing vector—a simple dilatation—which does not preserve the position of the two boundaries. As usual, the conformal transformation is uplifted as a diffeomorphism in AdS. And indeed, starting from the metric (5.7) with $\theta \in [0, 2\pi\gamma)$, we can apply the following coordinate transformation:

$$t_g \to \gamma\, t_g \ , \quad r \to r/\gamma \ , \quad \theta \to \gamma\theta \ , \tag{5.28}$$

after which the metric is still of the form (5.7), but with $\theta \in [0, 2\pi)$ and $q \to \tilde{q} = q\gamma^2$. In the latter coordinate system, the brane stretches between $\theta = 0$ and $\theta = -2\pi + \frac{\pi}{\sqrt{-\tilde{q}}}$. In turn, the portion of bulk geometry included in the system is identical to the case without the brane, and in particular the conical deficit is related to $\tilde{q}$ in the usual way: $2\pi(1 - \delta)$ with $\delta = (1 - \sqrt{-\tilde{q}})$. The transition to the BTZ black hole, which of course still happens as $\gamma \to \infty$, is signalled in this frame by the brane folding onto itself, and excising the whole conformal boundary. It is also useful to notice that the mass is $M = 4R^2(\sqrt{-\tilde{q}} + \tilde{q})$ and $\tilde{q}$ is defined in the interval $\left[-1, -\frac{1}{4}\right)$. In figure 13, we show the configuration of the brane for both positive and negative tension of the EoW brane.

### 5.2.2 $q > 0$: the BTZ black brane

In this case, the bulk geometry before the insertion of the brane is the one of a BTZ black brane [69]. In turn, the derivative in eq. (5.21) never diverges. Therefore, there is no turning point, and the brane intersects the horizon of the BTZ black brane at $r = \sqrt{q}L$ and goes all the way until it reaches the singularity. Since we want to describe the dual of a CFT on a strip, we then need two branes, ending on the conformal boundary at $\theta = 0$ and $\theta = -\pi$. Both of them cross the horizon and excise part of the black brane. Therefore there is no reason to make the coordinate $\theta$ periodic, which is why the local boundary operator is dual to a black brane rather than a black hole, see figure 14. The trajectories of

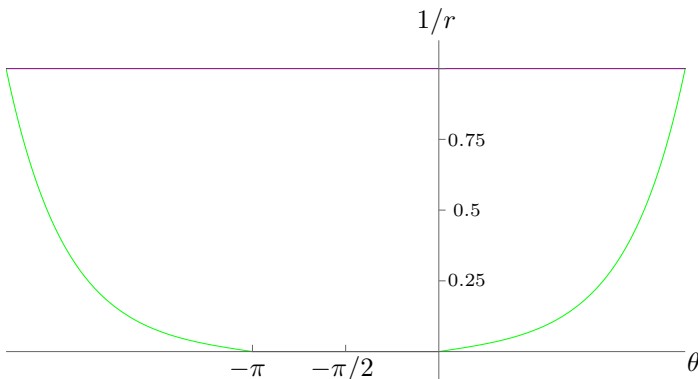

**Figure 14**: A constant $t_g$ slice of the black brane configuration. The EoW branes (in green) depart from the AdS boundary (in black) moving away from each other in the AdS bulk and hitting the black brane horizon (in purple).

the branes are obtained solving eq. (5.21) and then exploiting the symmetry $\theta \leftrightarrow -\pi - \theta$ [16]:

$$r_{\mathrm{BH}}^{(1)}(\theta) = \frac{\sqrt{q}L\tan\Theta}{\sinh(\sqrt{q}\,\theta)} \ , \qquad r_{\mathrm{BH}}^{(2)}(\theta) = -\frac{\sqrt{q}L\tan\Theta}{\sinh[\sqrt{q}(\theta+\pi)]} \ , \tag{5.29}$$

or

$$\theta_{\mathrm{BH}}^{(1)}(r) = \frac{1}{\sqrt{q}}\log\left(\sqrt{1+\left(\frac{\sqrt{q}L\tan\Theta}{r}\right)^2} + \frac{\sqrt{q}L\tan\Theta}{r}\right) \tag{5.30}$$

$$\theta_{\mathrm{BH}}^{(2)}(r) = -\pi - \frac{1}{\sqrt{q}}\log\left(\sqrt{1+\left(\frac{\sqrt{q}L\tan\Theta}{r}\right)^2} + \frac{\sqrt{q}L\tan\Theta}{r}\right) \ . \tag{5.31}$$

From the explicit expression (5.22) of the $(r,r)$ component of the metric, we see that, for $q > 0$, $r = r_H = \sqrt{q}L$ is an event horizon also on the EoW brane.

Finally, notice that the solution is well defined also in the interior, in the sense that the two branes do not meet before the singularity. This can be seen by going to Eddington-Finkelstein coordinates

$$\mathrm{d}s^2 = -(r^2 - qR^2)\,\mathrm{d}v^2 + 2R\,\mathrm{d}r\,\mathrm{d}v + r^2\mathrm{d}\theta^2 \ . \tag{5.32}$$

In this coordinates we can parametrise the EoW branes in terms of the coordinates $(v, r)$ and the matching condition will still be (5.21). Then, the solutions (5.29) are valid beyond the horizon in these coordinates and the branes do not intersect the horizon.

The same computation as in the previous subsection shows that the boundary operator dual to this state in AdS has scaling dimension given by eq. (5.27).

### 5.2.3 A puzzle at negative tension

Let us now briefly re-discuss the previous geometries, in the case where $\lambda < 0$. When $q < 0$, the main difference is that now the conical singularity is excised—see figure 13. Indeed, the solution respecting the boundary condition (5.17) is still eq. (5.23), but in this case $r > 0$ requires $\theta \leq 0$. The brane returns to the boundary at $\theta = -\pi/\sqrt{-q}$. Since point $r = 0$ does not belong to the spacetime, the

coordinate $\theta$ is not periodic. We can still apply the change of coordinate (5.28), in order to adjust the size of the strip. This time, $\gamma = 1/\sqrt{-q}$. But, as explained in subsection (5.2.1), this reveals the spacetime to simply be a slice of empty AdS, since $\tilde{q} = -1$. We conclude that this geometry is *not* dual to a heavy boundary operator, rather it produces the vacuum in the Poincaré patch.

The situation does not improve when considering the $q > 0$ case. Indeed, in this case we can notice that both the EoW branes (5.29) are defined in the interval $\theta \in [0, \pi]$ and they intersect non-smoothly at

$$r^* = \frac{\sqrt{q}\,L\,|\tan\Theta|}{\sinh\frac{\pi\sqrt{q}}{2}} \ . \tag{5.33}$$

This value may be behind the horizon, depending on the value of the tension, but the intersection always happens before the singularity. Such solution does not make sense within the simple EFT (5.1) and we have to discard it. We conclude that these gravitational solutions are not dual to heavy operators, when the tension of the EoW brane is negative. It would be interesting to explore this case further.

## 5.3 The time evolution of the holographic entanglement entropy

Having described the holographic dual to our state, we can turn to computing the entanglement entropy of the radiation on the gravity side. Since the matching of the energy density in the previous subsection fixed $q$ in terms of the scaling dimension $\Delta$, there are no other tunable parameters. Therefore, we expect the late time result (3.26) to be recovered exactly: our gravity dual lacks a light scalar [5], so we expect the displacement to give the dominant contribution to the OPE.

For convenience, we will first compute the holographic entanglement entropy in the black brane regime, and we will obtain the case of the conical singularity by analytic continuation.

Extremal surfaces in the AdS$_3$ geometry are geodesics. We can parametrize the curve using an affine parameter $\zeta$ and, using the symmetries of spacetime, we find first order differential equations for the geodesics:

$$\frac{\mathrm{d}}{\mathrm{d}\zeta}\left(K_\mu \frac{\mathrm{d}x^\mu}{\mathrm{d}\zeta}\right) = 0 \ , \tag{5.34}$$

where $K_\mu$ is a Killing vector. Two out of the six Killing vectors are trivial in global coordinates, $K_1 = \partial_\tau$ and $K_2 = \partial_\theta$. This is enough to fix the geodesic equations, together with the normalisation condition $\frac{\mathrm{d}x^\mu}{\mathrm{d}\zeta}\frac{\mathrm{d}x_\mu}{\mathrm{d}\zeta} = L^2$.

Then we find

$$\begin{aligned}
\frac{\mathrm{d}}{\mathrm{d}\zeta}t_{\mathrm{RT}}(\zeta) &= \frac{E}{(r_{\mathrm{RT}}(\zeta)^2 - qL^2)} \ , \\
\frac{\mathrm{d}}{\mathrm{d}\zeta}\theta_{\mathrm{RT}}(\zeta) &= \frac{\ell}{r_{\mathrm{RT}}(\zeta)^2} \ , \\
\left(\frac{\mathrm{d}}{\mathrm{d}\zeta}r_{\mathrm{RT}}(\zeta)\right)^2 &= (r_{\mathrm{RT}}(\zeta)^2 - qL^2)\left(1 - \frac{\ell^2}{L^2\,r_{\mathrm{RT}}(\zeta)^2}\right) + \frac{E^2}{L^2} \ ,
\end{aligned} \tag{5.35}$$

where $x^\mu_{\mathrm{RT}}$ is the trajectory of the RT surface in global coordinates,[9] and $E$, $\ell$ are two integration constants, corresponding to the energy and the angular momentum of the geodesic respectively. It is

---

[9]We omitted the subscript $g$ for global time in this case, hoping that it leads to no confusion.

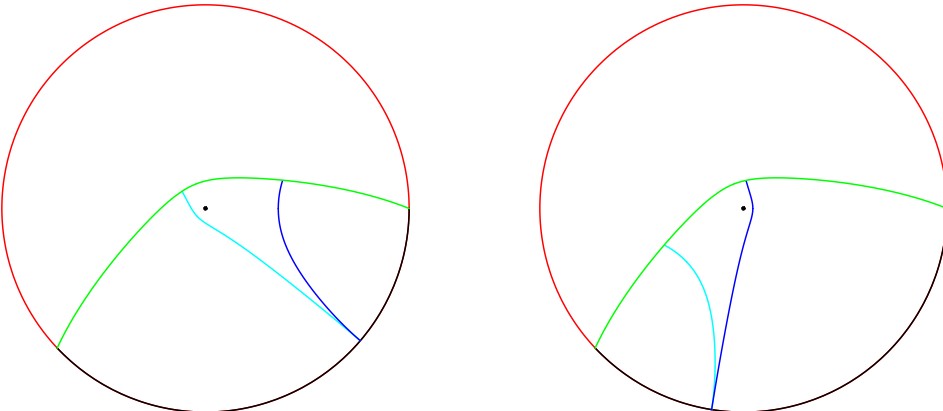

**Figure 15**: In general, there are two solutions for the orthogonality conditions, *i.e.* two curves which are local saddles of the geodesic length. We show these two geodesics (in blue and cyan) in the case of the conical singularity in the bulk. The brane is drawn in green. The left and right pictures differ by the location of the entangling surface: after the diffeomorphism (5.5) they will correspond to early and late time respectively. At the middle point of the strip, $\theta_\infty = -\pi + \frac{\pi}{2\sqrt{-\tilde{q}}}$, the two geodesics have the same length, then they exchange dominance: the blue one dominates at early time, the cyan one at late time. The case of the black brane is analogous: the two saddles end on the two EoW branes, respectively. Both these curves never hit the black brane horizon as the have a turning point at $r^* = L\sqrt{q \coth(\sqrt{q}\,\theta_\infty)} > r_H$.

convenient to parametrise the geodesics in terms of the coordinate $r$:

$$\frac{\mathrm{d}}{\mathrm{d}r}t_{\mathrm{RT}}(r) = -\frac{E\,L\,r}{(r^2 - qL^2)\sqrt{(r^2 - qL^2)(L^2\,r^2 - \ell^2) + E^2}} \ ,$$

$$\frac{\mathrm{d}}{\mathrm{d}r}\theta_{\mathrm{RT}}(r) = -\frac{\ell L}{r\sqrt{(r^2 - qL^2)(L^2\,r^2 - \ell^2) + E^2}} \ . \tag{5.36}$$

In the computation of the RT surface, we are after the minimal length geodesic anchored to the AdS boundary at $(t_{g,\infty}, r_\infty, \theta_\infty)$ and ending on the EoW brane [15]—see [34] for a justification of this prescription. It can be shown that such a geodesic is orthogonal to the EoW brane at their intersection point (for a detailed proof see *e.g.* [34]). Such orthogonality conditions completely specify the geodesic which correspond to the RT surface, *i.e.* the integration constants $E$ and $\ell$. In particular, we find

$$E = 0 \ ,$$
$$\ell^2 = \bar{r}^2 L^2 \cos^2\Theta + qL^4\sin^2\Theta \ , \tag{5.37}$$

where $\bar{r}$ is the radial coordinate of the intersection point between the RT surface and the brane. Notice that with $E = 0$, the geodesic sits on a constant global time slice and, in the geodesic equation (5.36), the horizon becomes a turning point of the geodesic anchored to the conformal boundary. This means that the RT surface never enters the BH horizon, as suggested by the doubling trick and the

considerations in [70]. Assuming $-\frac{\pi}{2} \le \theta_\infty \le 0$ we can solve the orthogonality condition for $\bar{r}$

$$\bar{r} = \sqrt{q}L \sqrt{\frac{\coth^2\left(\sqrt{q}\,\theta_\infty\right)}{\cos^2\Theta} - \tan^2\Theta}\ , \tag{5.38}$$

$$\ell = -\sqrt{q}L^2 \coth\left(\sqrt{q}\,\theta_\infty\right)\ ,$$

such that $\ell^2 > qL^4$, and the RT surface encounters a turning point before hitting the horizon. We can also notice that such geodesic is exactly the same one would find in the case without the End-of-World brane, for a symmetric interval with respect to the origin [4]. The length of this curve in units of $4G_N$ gives the leading contribution to the holographic entanglement entropy:

$$S_{\mathrm{BH}} = \frac{L}{4G_N} \left[\log \frac{2r_\infty \sinh\left(-\sqrt{q}\,\theta_\infty\right)}{\sqrt{q}\,L} + \operatorname{arcsinh}\tan\Theta\right] \qquad -\frac{\pi}{2} \le \theta_\infty < 0\ . \tag{5.39}$$

As mentioned above, the entropy is half of what it would be without brane, plus the contribution of the *boundary entropy* $\operatorname{arcsinh}\tan\Theta$. The substitution $\theta_\infty \to -\pi - \theta_\infty$ gives the entanglement entropy for $-\pi \le \theta_\infty < -\frac{\pi}{2}$. The RT surface in the latter case ends on the opposite brane. Notice that since we are discussing the holographic dual of a pure state, the homology constraint does not apply.

In order to obtain the holographic entanglement entropy in the case of the conical singularity we can analytically continue such result to $q < 0$:

$$S_{\mathrm{con}} = \frac{L}{4G_N} \left[\log \frac{2r_\infty \sin\left(-\sqrt{-\tilde{q}}\,\theta_\infty\right)}{\sqrt{-\tilde{q}}\,L} + \operatorname{arcsinh}\tan\Theta\right]\ . \tag{5.40}$$

The analytic continuation brings us to the set-up shown in figure 13, with the EoW brane attached to the boundary at $\theta = -2\pi + \frac{\pi}{\sqrt{-\tilde{q}}}$ and $\theta = 0$. This solution dominates for $-\pi + \frac{\pi}{2\sqrt{-\tilde{q}}} \le \theta_\infty \le 0$, while the leading saddle for $-2\pi + \frac{\pi}{\sqrt{-\tilde{q}}} \le \theta_\infty \le -\pi + \frac{\pi}{2\sqrt{-\tilde{q}}}$ is obtained from $\theta_\infty \to 2\pi - \frac{\pi}{\sqrt{-\tilde{q}}} - \theta_\infty$. In figure 15, we show the two competing geodesics.

Now, we need to map the coordinates $(t_{g,\infty}, r_\infty, \theta_\infty)$ to the time evolution in the Poincaré patch and study the entanglement entropy in terms of $(t, x)$ coordinates in the CFT. Recall that, in order to use eq. (5.5), the CFT should be defined in the interval $\theta \in [-\pi, 0]$. For the conical singularity, this means first changing coordinates such that the periodicity is $\theta \sim \theta + 2\pi\gamma$—see subsection 5.2.1. Then, from (5.5) we find

$$\tan\theta_\infty = \frac{2x\epsilon}{t^2 - x^2 + \epsilon^2}\ , \tag{5.41}$$

where we have already set $z = 0$. At early times $\theta_\infty \sim -\pi$, while at late times $\theta_\infty \sim 0$. In figure 16, we show the time evolution at $\mathscr{I}^+$ of the holographic entanglement entropy in the two regimes presented above.

In order to match this result with the bound presented in Section 3.3, we need to subtract the vacuum entanglement entropy which is trivially obtained from (5.40) setting $q = -1$. In the following, we will express explicitly the result for the late time behaviour only, as the early time configuration is fully symmetric. In the case $q > 0$, for $\theta_\infty \sim 0$ we find

$$\Delta S_A \sim \frac{(1+q)L}{24G_N}\theta_\infty^2 = \frac{2}{3}\Delta\zeta\ , \tag{5.42}$$

where we employed the relations $q = \frac{M}{L^2} - 1$, $M = 16LG_N\Delta$ and $\theta_\infty^2 \sim \frac{4\epsilon^2 x^2}{t^4} \sim \zeta$. The result for the conical singularity is more subtle as we need to perform first the diffeomorphism (5.28) with

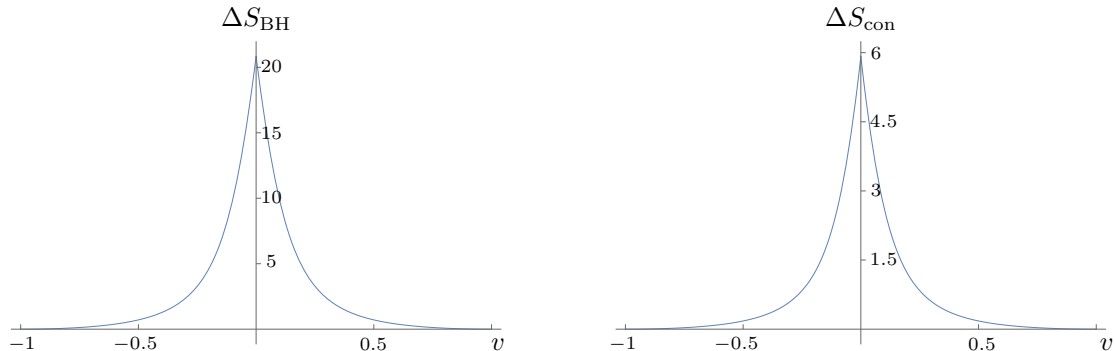

**Figure 16**: The time evolution on $\mathscr{I}^+$ of the holographic entanglement entropy in the case of the black brane and the conical singularity in the bulk. We have set $L = 1$, $G_N = 0.01$, $\epsilon = 0.3$, while $q = 1$ and $q = -0.5$ respectively in the two cases.

$\gamma^* = \frac{\sqrt{-\tilde{q}}}{2\sqrt{-\tilde{q}}-1}$ and the entanglement entropy becomes

$$\begin{aligned}
\Delta S_A &= \frac{L}{4G_N} \log \frac{\sin\left[(1-2\sqrt{-\tilde{q}})\,\theta'_\infty\right]}{(2\sqrt{-\tilde{q}}-1)\sin\left(-\theta'_\infty\right)} && -\frac{\pi}{2} \le \theta'_\infty \le 0 \\
&\sim \frac{(\sqrt{-\tilde{q}}+\tilde{q})L}{6G_N}\theta_\infty^2 = \frac{2}{3}\Delta\,\zeta && \theta'_\infty \sim 0\,,
\end{aligned} \tag{5.43}$$

where we used $M = 4(\sqrt{-\tilde{q}}+\tilde{q})L^2 = 16LG_N\Delta$ and $\theta_\infty^2 \sim \zeta$. Then we find that our bound (3.26) is perfectly saturated in both cases.

## 5.4 The island

As mentioned in the introduction, it is interesting to take a second look at the formulas of the previous subsection, in a different perspective. We integrate out the AdS bulk, and consider the 'bath+brane' system which includes the CFT and the EoW brane. It is expected that the island formula (1.1) can be used to recover eqs. (5.39) and (5.40). In fact, we can perform this computation explicitly in an expansion around the upper value for the tension—see eq. (5.20)—which corresponds to $\tan\Theta \to \infty$ [15, 34].

Let us first focus on the case of the black brane, eq. (5.39), for concreteness in the case where $-\pi/2 \le \theta_\infty < 0$. As the tension gets large, the two branes are pushed to the boundary, and the 'brane+bath' system approaches a CFT defined on the infinite line, at a uniform finite temperature, see figure 14. This description depends on the choice of quantization, and refers specifically to the time evolution generated by $t_g$. We will later comment on the time evolution generated by $t$. In order to compute the entanglement entropy, we are instructed to use eq. (1.1), looking for islands on the brane. Following [34], we notice that the area term is absent, since there is no Hilbert-Einstein term, nor a dilaton, in the effective action on the brane. For the entropy of the quantum fields, we can use the universal formula for a CFT at finite temperature, but we should be careful with the cutoff to be used for the entangling surface of the island. Calling for the moment $a$ and $a_I$ the cutoffs in the bath and on the brane respectively, the entanglement entropy obtained by tracing over an interval

$[-\theta_\infty, \theta_I]$, with $\theta_I > 0$ lying on the brane, is

$$S_{\text{gen}}(\theta, \theta_I) = \frac{c}{6} \log \frac{\left[ \frac{\beta}{\pi} \sinh \frac{\pi}{\beta}(\theta_\infty + \theta_I) \right]^2}{a \, a_I} \; , \tag{5.44}$$

where the subscript refers to the fact that this is the entropy to be inserted in eq. (1.1) before minimizing over the position of the island. We are choosing the boundary of the island to live on the right side of the bath, because we know where the dominating RT surface ends in this case: alternatively, the minimization procedure in the island formula would pick this case for us. The temperature $\beta$ can be derived from the black brane metric eq. (5.7) in the usual way:

$$\beta = \frac{2\pi}{\sqrt{q}} \; . \tag{5.45}$$

Let us come back to the cutoff. After writing the metric (5.7) in a Fefferman-Graham form, we see that the natural short distance cutoff is $a = L/r_\infty$, $r_\infty$ being the radial coordinate of the location of the CFT.[10] Now, we see from eq. (5.29) that the cutoff on the brane is position dependent, and for a point with coordinate $\theta_I$ is

$$a_I = \frac{\sqrt{q} \tan \Theta}{\sinh(\sqrt{q} \theta_I)} \; . \tag{5.46}$$

We are ready to write the island formula:

$$S_{\text{island}}(\theta_\infty) = \min_{\theta_I} S_{\text{gen}}(\theta_\infty, \theta_I) = \frac{c}{6} \left\{ \min_{\theta_I} \log \frac{\left[ \sinh \frac{\sqrt{q}}{2}(\theta_\infty + \theta_I) \right]^2}{\sinh \sqrt{q} \, \theta_I} + \log \frac{4 r_\infty \tan \Theta}{\sqrt{q} L} \right\} \; . \tag{5.47}$$

The minimization procedure yields $\theta_I = \theta_\infty$, which pleasingly puts the boundary of the island precisely where the RT surface ends on the brane. Replacing this value in eq. (5.47), we find

$$S_{\text{island}}(\theta_\infty) = \frac{c}{6} \left[ \log \left( \frac{2 r_\infty \sinh(-\sqrt{q} \theta_\infty)}{\sqrt{q} L} \right) + \log(2 \tan \Theta) \right] \; . \tag{5.48}$$

Comparing with eq. (5.39), we find a perfect match up to terms which vanish as $\tan \Theta \to \infty$. This is analogous to what was found in [34], and lends further support to the island formula, and to the matching procedure presented there and here. The matching of eq. (5.40) works in the same way, but now the 'bath+brane' system is placed in the pure state dual to the conical singularity discussed in [4]: notice that, again, the boundary is absent.

We conclude with some comments on the interpretation of the results of this subsection in the $(x, t)$ coordinates of our original setup, figure 1. A technical difficulty is given by the fact that the coordinate transformation (5.5) is periodic in $\theta$ with periodicity $2\pi$, while both solutions with the black brane and the conical singularity have a different periodicity—absent in the former case, fixed as explained in subsection 5.2.1 in the latter case. Therefore, the change of coordinates (5.5) is not invertible in a full constant $t_g$ slice of the bulk geometry, and therefore cannot be used to describe the whole brane. Instead of entering these details, we can use the fact that, as explained, we know the location of the island from the position of the RT surface. This allows to discuss its role in unitarizing

---

[10]In fact, the precise factor in the relation between the UV cutoff in the CFT and the position of the cutoff surface $r_\infty$ is immaterial. What matters is that we can only choose this convention once, then we must use it for both the entangling surface in the CFT and on the brane.

the radiation. Let us focus on the case of the black brane. At early time, the island lies on the left brane in figure 14, *i.e. beyond* the black hole from the point of view of the CFT degrees of freedom. The entanglement wedge of the radiation then does not include the black hole degrees of freedom.[11] After the Page time, on the other hand, the island includes the whole black hole, which becomes part of the entanglement wedge of the radiation. Notice that the island never falls behind the horizon, rather it jumps from one brane to the other at the Page time.

## 5.5 Energy measurements and local thermalization

In the previous subsection, we have seen that a sufficiently heavy boundary state creates an approximately thermal state when seen in global coordinates. What about the time dependent frame of figure 1? In this subsection, we analyse the energy distribution of the radiation, and point out that a large set of measurements indeed cannot distinguish between the radiation coming from a pure state and uncorrelated thermal radiation. Nevertheless, the latter model fails at describing the evolution of the entanglement entropy, and we must conclude that the detailed form of the correlations present between quanta emitted with short time separation is important.

If we compute the expectation value of multiple insertions of the stress tensor in a state created by a heavy boundary operator $\Delta/c \gg 0$, we find that it factorises

$$\langle \mathcal{O} | T(z_1) \dots T(z_n) | \mathcal{O} \rangle = \prod_{i=1}^{n} \langle \mathcal{O} | T(z_i) | \mathcal{O} \rangle \left[ 1 + \mathcal{O} \left( \frac{c}{\Delta} \right) \right] . \tag{5.49}$$

This behaviour is analogous to the factorization of multiple stress tensor correlators in a thermal state at large $c$

$$\langle T(\zeta_1) \dots T(\zeta_n) \rangle_\beta = \prod_{i=1}^{n} \langle T(\zeta_i) \rangle_\beta \left[ 1 + \mathcal{O}(c^{-1}) \right] . \tag{5.50}$$

Here, $\zeta_i$ are coordinates on the cylinder, but the right-hand side is constant at leading order at large $c$. While in (5.50) the leading (factorised) contribution comes from the Schwarzian in the map from the plane to the cylinder, in (5.49) the leading term comes from the first term in the recursion relation (A.6). Given this analogy, we are tempted to associate a temperature to the radiation of a heavy state [71]. Let us consider the one-point function in the two states. The thermal result is simply

$$\langle T(\zeta) \rangle_\beta = -\frac{c\pi^2}{6\beta^2} , \qquad \langle \bar{T}(\bar{\zeta}) \rangle_{\bar{\beta}} = -\frac{c\pi^2}{6\bar{\beta}^2} , \tag{5.51}$$

where in general the left and right temperatures can be different. For the local quench, instead, we have already encountered this correlator in section 3 and the result is

$$\frac{\langle \mathcal{O}(y_1) \mathcal{O}(y_2) T(z) \rangle}{\langle \mathcal{O}(y_1) \mathcal{O}(y_2) \rangle} = \frac{\Delta (y_1 - y_2)^2}{(z - y_1)^2 (z - y_2)^2} , \tag{5.52}$$

with a similar form for the antiholomorphic component. If we insist in defining a temperature, the radiation can almost look locally thermal, with

$$\frac{1}{\beta(t, x)} = \sqrt{\frac{6\Delta}{\pi^2 c}} \frac{\epsilon}{(x - t)^2 + \epsilon^2} , \qquad \frac{1}{\bar{\beta}(t, x)} = \sqrt{\frac{6\Delta}{\pi^2 c}} \frac{\epsilon}{(x + t)^2 + \epsilon^2} . \tag{5.53}$$

---

[11]In fact, the island *extends* the entanglement wedge of the radiation onto the left brane. Let us mention a technical detail: one might think of the entropy we computed as coming from a finite region of the CFT, in the limit where this region extends to infinity away from the boundary. Then a second twist operator, and the associated RT surface, appear. The latter always ends on the left brane, and is close to the AdS boundary. In the limit, it is insensitive to the presence of a non-trivial state, and its contribution is subtracted away in the difference with the vacuum, eq. (3.7).

Focusing on $\bar{\beta}$, we see that the temperature defined this way increases up to the Page time, then starts decreasing and vanishes in the late time limit. It is important to stress that the subleading terms in (5.50) do not match, and this identification makes sense only in the limit under consideration. Notice in particular that, if we bring the stress tensor insertions in eqs. (5.49) and (5.50) close enough to each other, the connected contributions, which are subleading in $c$ and $\Delta/c$, eventually dominate.

We can test whether the detailed form of these correlations is important by computing the (coarse grained) entropy under the assumption of collecting uncorrelated thermal radiation, and compare it with the entanglement entropy computed in the previous sections. This means essentially using the second law of thermodynamics for the radiation received in a short interval of time $dt$:

$$\mathrm{d}s = \bar{\beta}(t, x) \,\mathrm{d}e \ , \tag{5.54}$$

where $s$ is the entropy density and $e$ is the energy density. We denoted the temperature with $\bar{\beta}$ because we are interested in the thermodynamic entropy at $\mathscr{I}^+$, which is only reached by left moving modes, see eq. (5.53). Hence,

$$e(t, x) = \frac{\pi c}{6} \left( \bar{\beta}(t, x) \right)^{-2} \ . \tag{5.55}$$

Combining (5.54) and (5.55), we express the entropy density as function of the temperature:

$$s(t, x) = \frac{\pi c}{3} \left( \bar{\beta}(t, x) \right)^{-1} + s_0 \ , \tag{5.56}$$

with $s_0$ to be fixed. In particular, we choose $s_0 = 0$ by requiring that $s(0, x) \to 0$ as $x \to -\infty$. Integrating (5.56) along the space-like interval shown in figure 5, and disregarding the left-movers, we find the time evolution of the the entropy of a space-like interval $(-\infty, x_0]$ at fixed $t$[12]

$$\Delta S_{(-\infty, x_0]} = \int_\gamma s = \frac{\pi^2 c}{3} \int_{-\infty}^{x_0} \frac{\mathrm{d}x}{\bar{\beta}(t, x)} = \sqrt{\frac{c\Delta}{6}} \left( \pi + 2 \arctan \frac{x_0 + t}{\epsilon} \right) \ . \tag{5.57}$$

If we consider the limit of a light-like interval on $\mathcal{I}^+$ we find

$$\Delta S_{\mathcal{I}^+} = \sqrt{\frac{c\Delta}{6}} \left[ 2 \,\mathrm{arccot} \left( \epsilon \cot \frac{\pi V}{2} \right) + \pi \right] \simeq \sqrt{\frac{c\Delta}{6}} \,\pi\epsilon(V + 1) + \mathcal{O}(V + 1)^2 \ . \tag{5.58}$$

In particular, choosing $\epsilon = 1$, which is the same as picking $\ell = \epsilon$ in the change of coordinates (2.8), the right hand side is exact. The thermodynamic entropy grows at all times, which is not surprising since it is a coarse grained entropy, which assumes that no correlations exist between the quanta of radiation. In this sense, the result (5.57) reproduces the information paradox. However, notice that even at early times eq. (5.58) does not approximate the fine grained entropy. Indeed, it grows faster than the bound (3.37), that goes as $(V + 1)^2$. This means that, even if the correlators of the stress tensor, at leading order in $\Delta/c$ are compatible with the thermal answer, the density matrix of the state (2.1) is different from a locally thermal configuration at all times.

## 6 Outlook

In this work, we studied general properties of the entanglement of the radiation emitted from a boundary quench in a CFT, both in general and in a specific two-dimensional holographic setup. The region

---

[12]Notice that eq. (5.57) has the expected conformal properties: had we integrated on an interval $[\bar{z}_1, \bar{z}_2]$, with $\bar{z} = x + t$, it would have been a function of the cross ratio constructed from the two endpoints and the positions $y_1$, $y_2$ of the boundary operators in eq. (5.52). Since we fixed $y_1$, $y_2$ and an extremum of the interval, only dilatations are manifest in eq. (5.57).

of convergence of the OPE for the correlator (2.3), described in subsection 2.1, is a completely general result, valid in general dimension and for any choice of quasi-primary operators. Similarly, the entropy bound (3.37) is insensitive to any detail beyond the spacetime dimension, and its higher dimensional counterpart (3.52) depends on a single assumption about the boundary OPE of the stress tensor, and can be easily refined if this assumption is dropped. On the holographic side, we provided a description of the geometries dual to typical heavy boundary states, and computed both the entanglement entropy and the energy density of the radiation. We also showed that the island formula (1.1) can be derived from the RT prescription in our setup, in the large tension limit.

Various future directions are open. It would be interesting to sharpen the entropy bound in $d > 2$ by studying subleading terms in the OPE of the stress tensor, and apply it to specific theories, where the direct computation of the entanglement entropy might be too hard. Some of these theories might be holographic. In this context, one might enlarge the space of the bulk quantum fields and allow for the presence of a light scalar [5]. In the pure gravity setup, it would be interesting to solve the puzzle of the spectrum of heavy states supported by a brane with negative tension. Notice that the spectrum of the EoW brane is amenable to a conformal bootstrap study [72]: one might hope to sharpen this way our understanding of both positive and negative tension branes. A less ambitious task is to make precise the 'brane+bath' description of the states considered in section 5, in the time dependent frame where the bath lives on the semi-infinite line.

It is also important to think about generalizations of our setup. In general, the state of the radiation might be probed in more detail via higher point functions, which also come equipped with a richer set of OPE channels. More detailed information on the correlations present in the radiation might be obtained this way. On the holographic side, although the states considered here present black hole horizons on the EoW brane in a time dependent scenario, these black holes do not evaporate in one Poincaré time, and the Page curve of the radiation drops to zero not because of evaporation, but rather because the massive object in the bulk disappears behind the Poincaré horizon. It would be exciting to construct pure states which collapse into black holes [73], and couple them to a bath extended along the whole AdS boundary in global coordinates, so as to follow their evaporation process. Presumably, this will also require considering more general boundary conditions for the bath CFT, since a mass scale is necessary to create a long lived bound state on the boundary, which should be dual to a slowly evaporating black hole. It is worth mentioning that, while a non-conformal boundary condition endows the boundary with a stress tensor, this is still not enough to obtain a massless graviton on the brane. Indeed, the very existence of a coupling with the bath imposes a violation of the conservation of the stress tensor, and makes the graviton massive [74]. Massive gravitons have been linked to the appearance of islands, [12, 75] and from this point of view our setup does not make an exception.

## Acknowledgments

We would like to thank Costas Bachas, Joao Penedones and Julian Sonner for useful discussions. The research of LB is funded through the MIUR program for young researchers "Rita Levi Montalcini". SDA is supported by the European Union's Horizon 2020 research and innovation programme under the Marie Sklodowska-Curie grant agreement No. 764850 "SAGEX". MM is supported by the Swiss National Science Foundation through the Ambizione grant number 193472.

# A    Recurrence relation for $T(z)$ in BCFTs

In two-dimensional CFT the correlation functions involving a stress-tensor is in principle computable using a well-know recurrence relation [57]

$$\langle T(\xi)\mathcal{O}_1(z_1, \bar{z}_1)\cdots\mathcal{O}_m(z_m, \bar{z}_m)\rangle = \sum_{i=1}^{m}\left[\frac{h_i}{(\xi - z_i)^2} + \frac{1}{\xi - z_i}\partial_{z_i}\right]\langle\mathcal{O}_1(z_1, \bar{z}_1)\cdots\mathcal{O}_m(z_m, \bar{z}_m)\rangle \ . \quad (A.1)$$

This formula is an immediate consequence of the conformal Ward identities

$$\begin{aligned}
\delta_{\epsilon,\bar{\epsilon}}\langle\mathcal{O}_1(z_1, \bar{z}_1)\cdots\mathcal{O}_m(z_m, \bar{z}_m)\rangle &= \frac{1}{2\pi i}\oint_C dz\,\epsilon(z)\langle T(z)\mathcal{O}_1(z_1, \bar{z}_1)\cdots\mathcal{O}_m(z_m, \bar{z}_m)\rangle \\
&\quad - \frac{1}{2\pi i}\oint_C d\bar{z}\,\bar{\epsilon}(\bar{z})\langle\overline{T}(\bar{z})\mathcal{O}_1(z_1, \bar{z}_1)\cdots\mathcal{O}_m(z_m, \bar{z}_m)\rangle \ ,
\end{aligned} \quad (A.2)$$

where $C$ is a clockwise contour enclosing all the points $(z_i, \bar{z}_i)$. In 2D CFTs $\epsilon(z)$ and $\bar{\epsilon}(\bar{z})$ can be chosen independently and Cauchy's theorem implies (A.1). If we consider 2D boundary CFTs with boundary, for example, of the real axis and all operators $\mathcal{O}_i(z_i, \bar{z}_i)$ defined in the upper half-place ($\mathrm{Im}z \geq 0$), only transformations for which $\epsilon(z)$ is real analytic ($\overline{\epsilon(z)} = \epsilon(\bar{z})$) are allowed, in order to preserve the geometry. We also have to impose that the flux of energy across the boundary is zero:

$$T(z) = \overline{T}(z) \qquad \mathrm{Im}z = 0 \ , \quad (A.3)$$

which is known are Cardy's boundary condition. This allowed to extend the definition of $T(z)$ into the lower half-plane as

$$T(z) = \overline{T}(z) \qquad \mathrm{Im}z < 0 \ . \quad (A.4)$$

The integral in the right-hand side of (A.2) can be written as

$$\begin{aligned}
\delta_{\epsilon,\bar{\epsilon}}\langle\mathcal{O}_1(z_1, \bar{z}_1)\cdots\mathcal{O}_m(z_m, \bar{z}_m)\rangle &= \frac{1}{2\pi i}\oint_C dz\,\epsilon(z)\langle T(z)\mathcal{O}_1(z_1, \bar{z}_1)\cdots\mathcal{O}_m(z_m, \bar{z}_m)\rangle \\
&\quad + \frac{1}{2\pi i}\oint_{\overline{C}} dz\,\epsilon(z)\langle T(z)\mathcal{O}_1(z_1, \bar{z}_1)\cdots\mathcal{O}_m(z_m, \bar{z}_m)\rangle \ ,
\end{aligned} \quad (A.5)$$

where $\overline{C}$ is a counter-clockwise contour on the lowest half place, enclosing all the points $\bar{z}_i$. If there are no boundary operator insertions the contours $C$ and $\overline{C}$ cancel on the boundary. From Cauchy's theorem, the recurrence relation for the correlation function with a generic number of stress-tensor with bulk operators in presence of a boundary becomes [47]:

$$\begin{aligned}
&\langle T(\xi)T(\xi_1)\cdots T(\xi_n)\mathcal{O}_1(z_1, \bar{z}_1)\cdots\mathcal{O}_m(z_m, \bar{z}_m)\rangle = \\
&\sum_{i=1}^{m}\left[\frac{h_i}{(\xi - z_i)^2} + \frac{\bar{h}_i}{(\xi - \bar{z}_i)^2} + \frac{1}{\xi - z_i}\partial_{z_i} + \frac{1}{\xi - \bar{z}_i}\partial_{\bar{z}_i}\right]\langle T(\xi_1)\cdots T(\xi_n)\mathcal{O}_1(z_1, \bar{z}_1)\cdots\mathcal{O}_m(z_m, \bar{z}_m)\rangle + \\
&\sum_{i=1}^{n}\left[\frac{2}{(\xi - \xi_i)^2} + \frac{1}{\xi - \xi_i}\partial_{\xi_i}\right]\langle T(\xi_1)\cdots T(\xi_n)\mathcal{O}_1(z_1, \bar{z}_1)\cdots\mathcal{O}_m(z_m, \bar{z}_m)\rangle + \\
&\sum_{i=1}^{n}\frac{c}{2(\xi - \xi_i)^4}\langle T(\xi_1)\cdots T(\xi_{i-1})T(\xi_{i+1})\cdots T(\xi_n)\mathcal{O}_1(z_1, \bar{z}_1)\cdots\mathcal{O}_m(z_m, \bar{z}_m)\rangle \ ,
\end{aligned}$$
$$(A.6)$$

where we included for completeness also the contribution of an arbitrary number of stress tensor in the correlation function.

The recurrence relation can be extended to boundary operators:

$$\langle T(\xi)\cdots\hat{\mathcal{O}}_k(y_k)\cdots\rangle = \left[\frac{\hat{\Delta}}{(\xi-y_k)^2} + \frac{1}{\xi-y_k}\partial_{y_k}\right]\langle\cdots\hat{\mathcal{O}}_k(y_k)\cdots\rangle + \cdots , \tag{A.7}$$

where $\hat{\Delta}$ is the scaling dimension of the boundary operator. This can be easily verified from (A.6) by considering a generic boundary OPE for a bulk operator

$$\mathcal{O}(z,\bar{z}) \sim \cdots + (z-\bar{z})^{\Delta-\hat{\Delta}}\hat{\mathcal{O}}(y) + \dots . \tag{A.8}$$

In our setup the boundary is defined on the imaginary axis $z = -\bar{z}$, then the formulas above can be applied after replacing $z \to -\bar{z}$.

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
