# Peer review of "Radiation, entanglement and islands from a boundary local quench"

_SciPost Physics_

## Round 1 · Referee Report · Anonymous (Referee 1) · 2022-9-3

Strengths

Well written, very clear and easy to follow. Interesting results.

Weaknesses

None, other than the small clarifications present in my report.

Report

See report attached.

Attachment

  • validity: top
  • significance: good
  • originality: good
  • clarity: top
  • formatting: perfect
  • grammar: perfect

Author:  Stefano De Angelis  on 2023-02-07  [id 3331]

(in reply to Report 1 on 2022-09-03)

We would like to thank the referee for a careful reading of our paper and for the insightful comments. We made some improvements in various parts of our work to address the points raised in the report.

  • The referee asked about the gravitational interpretation of the relative entropy bound. While addressing this question, we took the occasion to delve in more detail into the microscopic origin of the universal part of the entanglement entropy as well. We extended the discussion in sections 3.2, 3.3.1, 4.1, and 5.3. In Section 3.2 and 3.3.1, we show that the sl(2) block of the displacement operator coincides with the relative entropy bound. In section 4.1, after eq. 4.12, we refined the microscopic interpretation of the universal part of the entropy (i.e. the bound) as the total contribution of single copy operators in the replica orbifold. Incidentally, section 4.1 has been slightly reorganised: for additional clarity, we defined a conformal map that sends the upper half plane into itself (see the new figure 9) and rephrased the computation in terms of the latter. Finally, we come to the holographic interpretation of the bound at the end of section 5.3. The discussion after eq. 5.45 is new, including figure 16: there, we make closer contact with the previous results in the literature, and we point out that the bound acquires a clear holographic interpretation in the limit where the excitation above the vacuum becomes light.

  • We agree that the comment about the operator O being single-trace was confusing, and we have removed it.

  • We agree with the referee and point his/her attention to the last comment before section 5.1, which we added in the revised version. We also make the microscopic interpretation of the holographic entanglement entropy in terms of the Virasoro module of the identity completely precise, in section 5.3.

  • The inequality is correct. In this section, we consider Delta >> c, which is needed in equation 5.53 and makes possible the definition of the local temperature in terms of the CFT data (equation 5.57). We added a clarification at the beginning of the section.

  • The references suggested by the referee and few more others have been added.

  • We have also taken the chance to correct some typos in the text.

---

## Round 1 · Referee Report · Anonymous (Referee 2) · 2022-9-9

Strengths

See below

Weaknesses

See below (basically none!)

Report

This is a very well-written paper that discusses a topic of great current interest; it begins with a study from conformal field theory of the entanglement entropy during a quench in the presence of a boundary. In particular, the authors make use of 2d conformal invariance to both:
a) derive universal formulas for the EE at late times, given some assumptions on the defect operator spectrum, and
b) derive bounds on the growth of this quantity using general properties of the EE.

This result is of intrinsic interest to the EE-in-many-body-physics community. However the authors then go on to discuss the holographic dual; I found the discussion to be very complete and provide a wealth of detail on how to construct the solutions from different points of view. The authors find agreement with general CFT expectations; finally they also show that the result can be viewed as support for the “Island” prescription if one considers a duality frame where the bulk of AdS_{3} is considered in a field-theoretical duality frame, but the brane itself is considered in a dynamical gravity duality frame. (This agreement is expected along the lines of recent work in Refs [33, 34], but it is still nice to see in a slightly new kinematics).

I found this paper to be extremely well-written, timely, and providing a wealth of detail. Though none of its conclusions are very unexpected, the detailed computations and CFT results are quite valuable. I can find little to improve and recommend publication in its current form.

Requested changes

Nothing.

  • validity: top
  • significance: high
  • originality: high
  • clarity: top
  • formatting: perfect
  • grammar: perfect

Author:  Stefano De Angelis  on 2023-02-07  [id 3332]

(in reply to Report 2 on 2022-09-09)

We would like to thank the referee for their accurate assessment of our paper and for their positive evaluation.

---

## Editorial Decision

resubmitted